# The interplay between dietary fatty acids and gut microbiota influences host metabolism and hepatic steatosis

Marc Schoeler [1,18], Sandrine Ellero-Simatos [2,18], Till Birkner [3,4,18], Jordi Mayneris-Perxachs [5,6,7], Lisa Olsson [1], Harald Brolin[1], Ulrike Loeber [3,4,8,9], Jamie D. Kraft[1], Arnaud Polizzi [2], Marian Martí-Navas[10], Josep Puig[10], Antonio Moschetta [11,12,13], Alexandra Montagner[14], Pierre Gourdy[14,15], Christophe Heymes[14], Hervé Guillou[2], Valentina Tremaroli[1], José Manuel Fernández-Real [5,6,7,16], Sofia K. Forslund [3,4,8,9,17], Remy Burcelin [14] & Robert Caesar [1] ✉

Dietary lipids can affect metabolic health through gut microbiota-mediated mechanisms, but the influence of lipid-microbiota interaction on liver steatosis is largely unknown. We investigate the impact of dietary lipids on human gut microbiota composition and the effects of microbiota-lipid interactions on steatosis in male mice. In humans, low intake of saturated fatty acids (SFA) is associated with increased microbial diversity independent of fiber intake. In mice, poorly absorbed dietary long-chain SFA, particularly stearic acid, induce a shift in bile acid profile and improved metabolism and steatosis. These benefits are dependent on the gut microbiota, as they are transmitted by microbial transfer. Diets enriched in polyunsaturated fatty acids are protective against steatosis but have minor influence on the microbiota. In summary, we find that diets enriched in poorly absorbed long-chain SFA modulate gut microbiota profiles independent of fiber intake, and this interaction is relevant to improve metabolism and decrease liver steatosis.

Non-alcoholic fatty liver disease (NAFLD) is the most common chronic liver disease and a leading cause of liver-related mortality[1]. Hepatic steatosis, the first step in NAFLD development, is determined by several factors including uptake and disposal of fatty acids, de novo lipogenesis and fatty acid oxidation within the liver[2]. The composition of dietary fatty acids can affect several of these processes[3].

NAFLD is linked to an aberrant gut microbiota. Studies in mice have demonstrated that NAFLD-associated gut microbiota can contribute to the development of several metabolic perturbations[4,5]. In humans, microbiome signatures discriminating NAFLD patients from healthy individuals have been identified[6–8]. Furthermore, fecal microbiota transfer from patients with hepatic steatosis has been shown to increase hepatic fat content in recipient mice[7,9], supporting a causal relationship between the gut microbiota and NAFLD.

Interaction between dietary lipids and the gut microbiota can affect host physiology and disease development[10]. We have previously shown that differences in adiposity and adipose tissue inflammation between mice fed lard diet or fish oil diet are transmitted by gut microbiota transfer[11]. Similarly, milk fat has been shown to induce a pro-inflammatory immune response and colitis through a microbiota-mediated mechanism[12]. A milk fat diet has also been shown to induce steatosis[13,14]. However, it is still not known how the interaction between dietary lipids and the gut microbiota affects hepatic steatosis.

In the present paper we aim to determine how dietary fatty acid composition affects the gut microbiota profile, microbiota-mediated metabolic regulation, and development of hepatic steatosis.

## Results

### Dietary fatty acids affect the gut microbiota in humans

To investigate how consumption of dietary fats with different fatty acid compositions affects gut microbiota composition in humans, we divided subjects ($n = 117$) into tertiles (low, medium, and high consumers) based on their consumption of saturated fatty acids (SFA), monounsaturated fatty acids (MUFA) or polyunsaturated fatty acids (PUFA). Principal component analysis (PCA) on fecal microbiome data (Fig. 1a) showed a significant clustering of samples according to tertiles of SFA consumption ($p = 0.005$, permutational multivariate analysis of variance, 1000 permutations), near-to-significant ($p = 0.056$) clustering of samples according to tertiles of MUFA consumption, and significant ($p = 0.02$) clustering of samples according to tertiles of PUFA consumption. Subjects consuming low amounts of SFA had higher α-diversity, measured as Fisher's alpha (Fig. 1b) and observed microbial species (Supplementary Fig. 1a) but similar

Shannon index (Supplementary Fig. 1b) compared to those consuming higher amounts of SFA. Low, medium, and high consumers of MUFA and PUFA did not differ significantly in any of the measured α-diversity indexes (Fig. 1b and Supplementary Fig. 1a, b).

To identify bacterial taxa affected by the amounts of SFA, MUFA or PUFA consumed, we fitted robust linear regression models for each taxon controlling for age, body mass index (BMI), sex and dietary fiber intake. The abundance of several bacteria from phyla Firmicutes and Spirochaetes, including cellulolytic bacteria such as *Acetivibrio cellulolyticus*, *[Clostridium stercorarium]* and *[Clostridium] cellulosi*, was negatively associated with the amount of dietary SFA (Fig. 1c). Only eight bacteria were positively associated with high intake of SFA (Fig. 1c) and included Proteobacteria and bacteria from the oral cavity and skin. These results are in line with the effects on α-diversity (Fig. 1a and Supplementary Fig. 1a). Bacteria negatively associated with MUFA intake partly overlapped with those linked with SFA intake, but the number of correlations and the fold change difference were smaller (Supplementary Fig. 1c). Only two bacterial species were significantly associated with PUFA intake (Supplementary Fig. 1d).

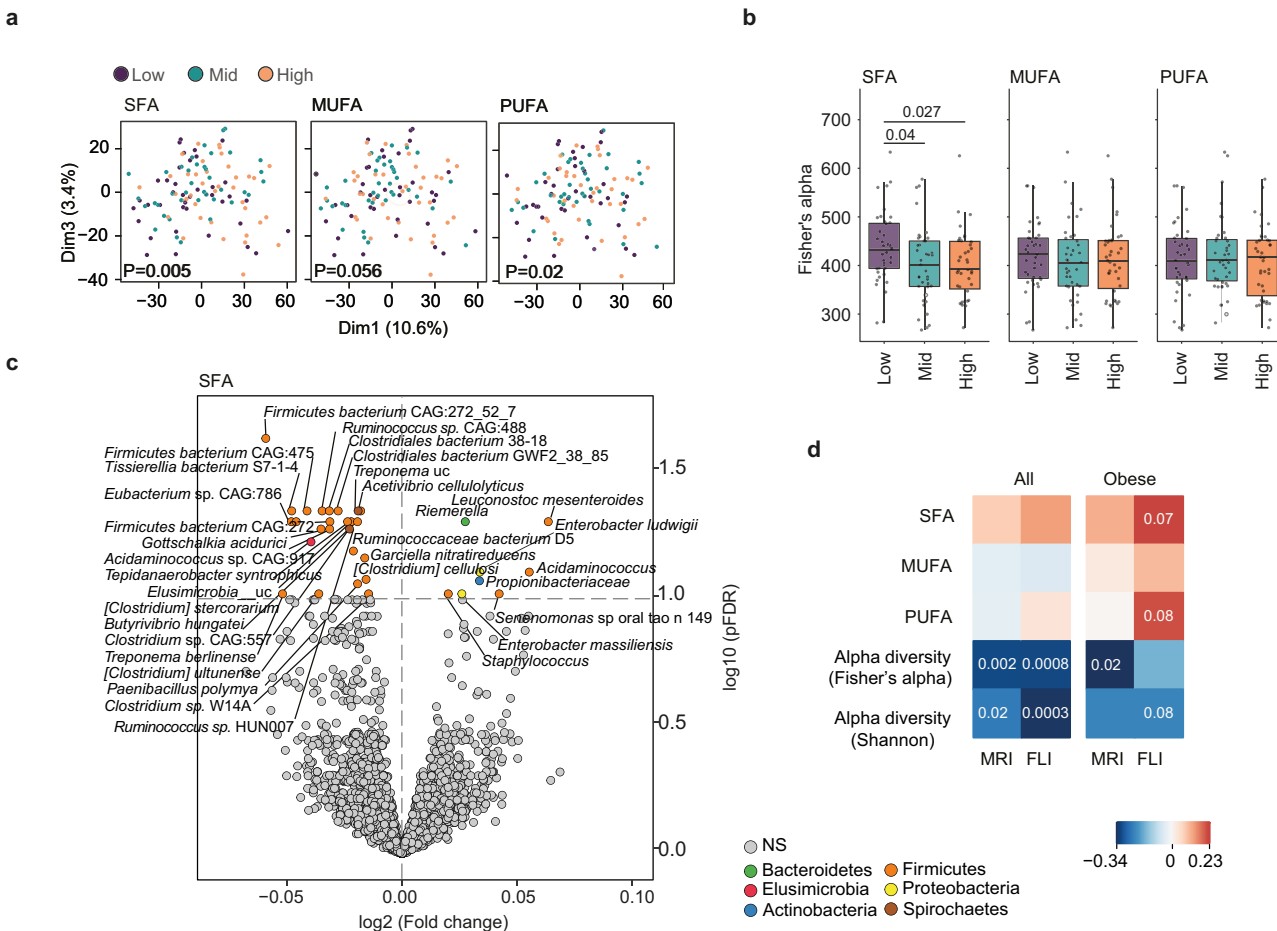

**Fig. 1 | Increased consumption of SFA decreases gut microbiota diversity and abundance of butyrate-producing bacteria in humans.** Subjects from the IRONMET cohort. **a** Principal component analysis (PCA) on fecal microbiome data according to tertiles of SFA, MUFA and PUFA consumption. **b** Boxplots of α-diversity measured by Fisher's alpha. Subjects were divided into tertiles based on SFA, MUFA or PUFA consumption. Each subject was included in the analysis of all three fatty acid categories ($n = 39$ per category and tertile). There were no significant differences in sequencing depth among tertiles. **c** Volcano plot of differential bacterial (pFDR <0.05) associated with the amount of SFA consumed identified after fitting a robust linear regression model to the clr-transformed data controlling for age, sex, BMI and fiber intake. Fold change associated with a unit change in SFA and FDR-adjusted

values (pFDR) are plotted for each taxon. Significant taxa are colored according to phylum as shown. **d** Spearman correlation of SAT, MUFA and PUFA consumption, alpha diversity (Fisher's alpha and Shannon diversity) against liver fat determined by MRI and FLI in all subjects, and in only obese subjects ($n = 85$) after controlling for age and sex. $n = 117$. Significant p-values as determined by two-sided Wilcoxon rank-sum test are displayed in the figure. For box plots in (**b**): the middle line is the median, the lower and upper hinges are the first and third quartiles, the whiskers extend from the hinge to the largest and smallest value no further than 1.5 × the interquartile range (IQR). SFA saturated fatty acids, MUFA mono-unsaturated fatty acids, PUFA poly-unsaturated fatty acids. See also Supplementary Fig. 1. Source data are provided as a Source data file.

To investigate the association between dietary lipids, gut microbiota diversity and liver fat we correlated dietary SFA, MUFA and PUFA, Fisher's alpha and Shannon index with liver fat determined by Magnetic Resonance Imaging (MRI) and Fatty Liver Index (FLI) in all individuals and in in subset including only individuals with obesity ($n = 85$) (Fig. 1d). The intake of SFA and PUFA correlated positively to FLI in obese individuals, but the correlation did not reach significance in the full cohort. Both Fisher's alpha and Shannon index were negatively correlated to liver fat determined by MRI and FLI in all individuals. The correlations between Fisher's alpha and MRI and between Shannon index and FLI were also significant in the subset including only individuals with obesity.

Taken together, we find that dietary fat affects gut microbiota profile in humans. SFA has a stronger effect on the gut microbiota than MUFA and PUFA, and low consumption of SFA is associated with higher gut microbiota α-diversity and higher abundance of specialized bacteria able to degrade dietary fibers. Moreover, we find a negative correlation between gut microbiota diversity and steatosis, and positive correlation between dietary SFA and MUFA and FLI in obese individuals. Our results also suggest that the effect of dietary fats is independent of the dietary fiber intake.

## Dietary fatty acid composition affects hepatic steatosis

To investigate the influence of dietary fatty acids on the gut microbiota and determine how this interaction modulates host metabolism, we fed mice eight isocaloric diets with the same fiber content but different lipid composition for 9 weeks (Fig. 2a, b). A milk fat (MF) diet that has previously been shown to induce steatosis[13,14] served as reference diet. MF contains high levels of palmitic acid that induces obesity, dyslipidemia and hepatocyte lipid accumulation[15]. In the other diets (termed diets A–G), different fat sources were combined to obtain specific shifts in fatty acid composition compared to MF (Fig. 2a, b). Diets A–C had similar amounts of SFA as MF but decreased levels of palmitic acid and increased levels of stearic acid (A and C) and/or medium-chain fatty acids (B and C). Diet D had lower levels of SFA and higher levels of MUFA, and diets E-G had lower levels of SFA and higher levels of *n-3* PUFA (diet E), *n-6* PUFA (diet F) or a combination of *n-3* and *n-6* PUFA (diet G) (Fig. 2b).

Mice fed diets A, B and C were leaner than mice fed MF diet (Fig. 2c and Supplementary Fig. 2a–d). Cage-wise weekly measurements of diet consumption indicated that decreased adiposity was not explained by reduced food intake (Supplementary Fig. 2e). Mice fed diets A and C had a trend towards increased fecal levels of cholesterol (Supplementary Fig. 2f) and increased levels of free fatty acids (Fig. 2d), which showed a major increase in the degree of saturation (Supplementary Fig. 2g). Mice fed diet B had increased fecal levels of cholesteryl esters (Supplementary Fig. 2h). Fecal levels of triglycerides did not differ between the dietary groups (Supplementary Fig. 2i). Diet A gave rise to significantly improved glucose tolerance compared with mice fed MF (Supplementary Fig. 2j-m). Strikingly, diet A, B, C, E, and G also gave rise to strongly reduced hepatic fat accumulation compared with mice fed MF (Fig. 2e–g and Supplementary Figs. 2n–o and 3).

## Dietary fatty acid composition affects lipid metabolism

To investigate the mechanisms underlying the observed differences in hepatic lipid levels, we performed microarray analysis on liver transcriptomes. PCA based on gene expression data showed that mice fed diets A and C grouped together, while mice fed diets E and G grouped together (Supplementary Fig. 4a). Both these groups separated from mice fed MF (A and C vs MF $p = 0.05$; E and G vs MF $p = 0.001$).

To identify biological processes that differed between diet-induced gene expression profiles we performed gene ontology (GO) enrichment analysis and transcription target enrichment analysis. Genes upregulated in mice fed diets A and C, and to a lesser extent diet B, were enriched in functional categories and targets for transcription

factors associated with cholesterol biosynthesis (e.g., SREBPF2 and NR5A2) (Supplementary Fig. 4b, c). Genes upregulated in mice fed diets E and G were enriched in functional categories and targets for transcription factors associated with fatty acid metabolism (e.g., HNF4a and HNF1a) (Supplementary Fig. 4b, c). Genes downregulated in mice fed diets E and G were enriched in functional categories and targets for transcription factors associated with fatty acid biosynthesis (e.g., SREBP1 and ChREBP) (Supplementary Fig. 4d, e). Genes downregulated in mice fed diets D and F, as well as in mice fed the other diets, were enriched in functional categories and targets for transcription factors associated with immune functions compared to mice fed MF diet (Supplementary Fig. 4d, e). As expected from the observed differences in steatosis (Fig. 2e–g), targets for several transcription factors associated with steatosis[16] were downregulated in most diets compared to the MF diet (Supplementary Fig. 4d, e).

At the individual transcript level, we found that expression of *Hmgcs1* (3-hydroxy-3-methylglutaryl-Coenzyme A synthase 1), a rate-limiting enzyme in cholesterol biosynthesis, was upregulated in mice fed diets A and C (Fig. 2h), while the expression of *Fasn* (fatty acid synthase), a rate-limiting enzyme in de novo lipogenesis, was downregulated in mice fed diets E, F, and G (Fig. 2i). Sparse partial least squares (sPLS) regression analysis between hepatic mRNA expression and hepatic triglyceride content showed that mice fed diets E and G separated from MF on the first dimension while mice fed A, C, and B separated from MF on the second dimension (Fig. 2j). Functional categories related to the first dimension included fatty acid and triglyceride biosynthesis, while functional categories related to the second dimension included sterol and cholesterol metabolism (Supplementary Table 1).

We investigated if the observed differences in expression of genes involved in inflammatory processes were reflected in hepatic immune cell infiltration but did not observe significant differences between the dietary groups (Supplementary Fig. 4f). We also measured expression of genes encoding the tight junction proteins zonulin (*Hp*) and occludin (*Ocln*) in ileum (Supplementary Fig. 4g, h) and serum abundance of cytokines but did not observe significant differences between the groups except for small increases in interferon-γ and interleukin (IL)-4 in mice fed diets G and C, respectively (Supplementary Table 2).

Taken together, diet A and C, enriched in stearic acid, gave rise to increased expression of genes involved in hepatic cholesterol biosynthesis while diet E, F and G, enriched in PUFA, gave rise to decreased expression of genes involved in hepatic de novo lipogenesis.

## Dietary fatty acid composition affects the gut microbiota

Next, we profiled the cecal microbiota of mice fed the eight diets by 16S rRNA gene sequencing. Principal coordinate analysis (PCoA) of Bray–Curtis dissimilarity showed significant grouping of samples according to diet (Fig. 3a and Supplementary Table 3), and permutational multivariate analysis of variance (adonis, 9999 permutations) showed that fat source explained 45% of the variability in microbiota composition ($R^2 = 0.45$, $p = 0.0001$). Pairwise analysis of similarity (ANOSIM, 9999 permutations) showed that all diet comparisons, except for MF vs D, D vs F, and E vs G, differed significantly in the means of ranked Bray-Curtis dissimilarity (Supplementary Table 3). Microbiota from mice fed diet C and diet A differed the most from mice fed MF diet ($R^2 = 0.87$ and $R^2 = 0.73$, respectively; Supplementary Table 3). Pairwise analysis of similarity on unweighted UniFrac data showed significant differences between all diets (Fig. 3b and Supplementary Table 3).

The microbiota of mice fed diets A and C had fewer observed species and lower phylogenetic diversity than mice fed MF (Fig. 3c and Supplementary Fig. 5a), but no significant reduction in bacterial load (Fig. 3d), suggesting that dietary fats in diets A and C affected specific microbial taxa but not the cellular density of the communities. This

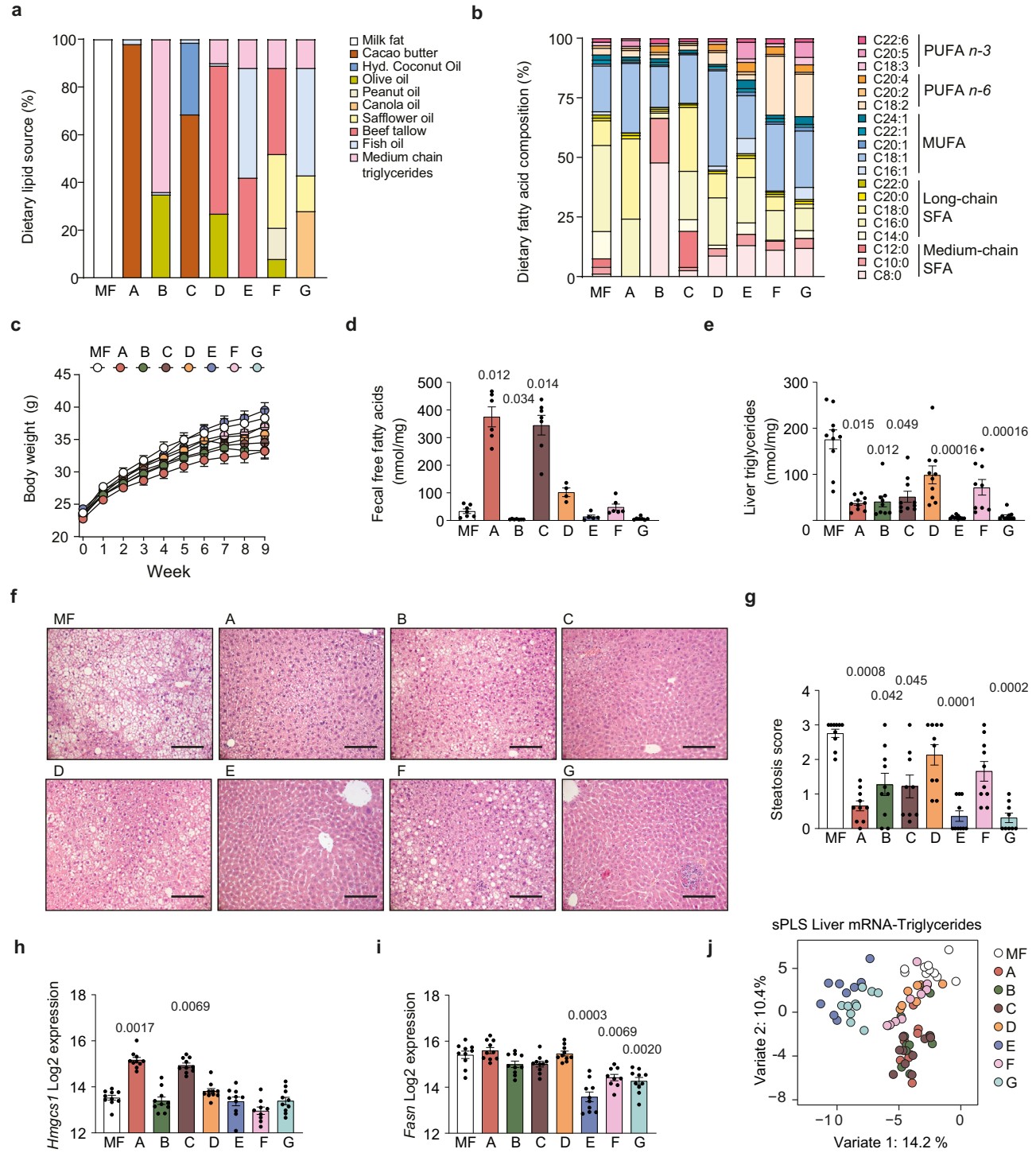

**Fig. 2 | Dietary fatty acid composition influences adiposity, steatosis, fecal lipid abundance and hepatic gene expression in mice.** Eight groups of mice were fed high-fat diets with different lipid compositions (MF, A–G) for 9 weeks. **a** Fat sources and **b** fatty acid composition of diets. **c** Body weight, **d** fecal concentration of free fatty acids, **e** liver triglyceride concentration, **f** hematoxylin and eosin staining of liver tissue (scale bar = 100 μm) and **g** steatosis score. Hepatic expression of **h** *Hmgcs* and **i** *Fasn* determined by microarray analysis. **j** Sparse partial least squares regression (sPLS) analysis of microarray data and liver triglyceride levels liver triglyceride concentration. **c**, **g**–**j**: *n* = 10 except for F where *n* = 9; **d**: *n* = 7 (MF), 6 (A), 7 (B), 7 (C), 4 (D), 5 (E), 6 (F), and 7 (G); **e**: *n* = 10 except for B and F where *n* = 9. Significant *p* values vs MF diet as determined by two-sided Kruskal–Wallis test are displayed in (**d**–**i**). Data are presented as mean ± SEM. SFA saturated fatty acids, MUFA mono-unsaturated fatty acids, PUFA poly-unsaturated fatty acids, *Hmgcs1* 3-hydroxy-3-methylglutaryl-Coenzyme A synthase 1, *Fasn* fatty acid synthase. See also Supplementary Figs. 2 and 3 and Supplementary Table 1. Source data are provided as a Source data file.

observation was confirmed by the analysis of differentially abundant zero-radius operational taxonomic units (zOTUs), which showed that mice fed diets A and C had similarly altered zOTU proportions both at the phylum (Fig. 3e) and genus (Fig. 3f) level compared with mice on MF. Examples of these changes included a higher proportion of the mucus degraders and propionate producers *Akkermansia* and *Bacteroides* and a lower proportion of several butyrate producers (e.g., *Roseburia*, *Oscillibacter*, *Anaerotruncus*, and *Intestinimonas*) (Fig. 3f).

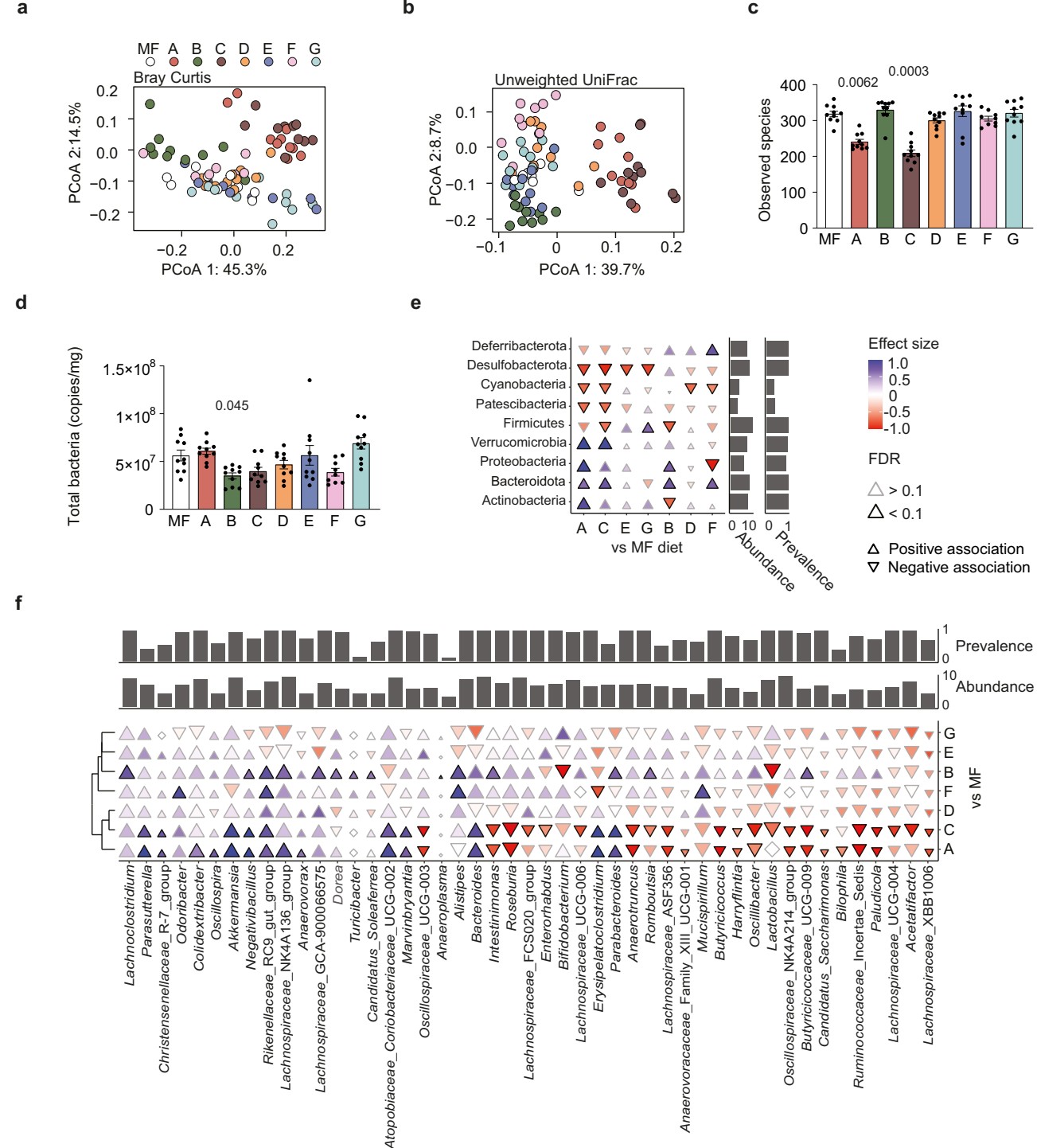

**Fig. 3 | Dietary fatty acid composition influences cecal microbiota profile in mice.** Analysis of cecal microbiota by 16S rRNA gene sequencing in eight groups of mice fed high-fat diets (MF, A-G) for 9 weeks. Principal coordinate analysis (PCoA) based on **a** Bray–Curtis dissimilarity and **b** unweighted UniFrac. **c** Number of observed species. **d** Quantification of total bacterial counts in cecum by qPCR. Differentially abundant **e** phyla and **f** genera in mice on diets A-G compared to MF diet, controlled for cage effects. Black-bordered triangles in panel E and F: Two-sided Wilcoxon rank sum test FDR-adjusted $p$ value < 0.1. The direction and coloring of triangles indicate Cliff's Delta effect size. The triangle size and left (**e**)/ lower (**f**) marginal bar plot depict log transformed abundance of individual genera. Right (**e**)/upper (**f**) marginal bar plot shows proportion of samples containing respective genera. Source data are provided as a Source data file. $n = 10$ except for F where $n = 9$. Significant $p$ values vs MF diet as determined by two-sided Wilcoxon rank sum test are displayed in the figure. Data are presented as mean ± SEM. See also Supplementary Fig. 5 and Supplementary Table 3. Source data are provided as a Source data file.

The gut microbiota of mice fed diet B was distinct from that of those fed the other tested diets, with decreased levels of *Lactobacillus* and *Bifidobacterium* and increased levels of the butyrate producers *Alistipes*, *Butyricicoccaceae*, and *Intestinimonas* compared to mice fed MF.

The difference in gut microbiota profiles between mice fed diet C and mice fed MF were paralleled by altered cecal levels of short-chain fatty acids (SCFA), namely decreased levels of butyrate, increased levels of succinate and a non-significant increase in propionate levels

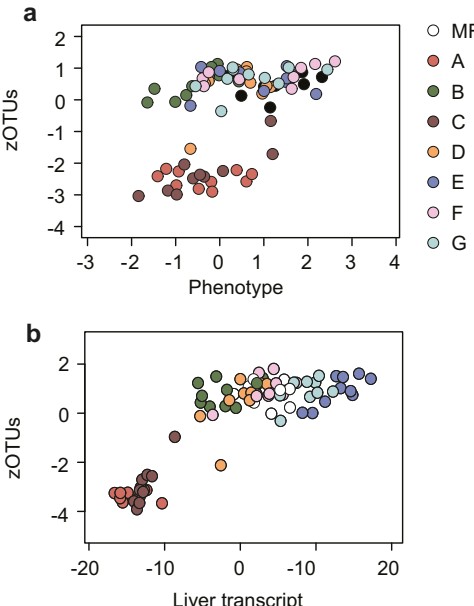

**Fig. 4 | Covariance between cecum microbiota, host metabolic features and hepatic gene expression.** **a** N-integration discriminant analysis with DIABLO between zOTUs in the cecal microbiota and host metabolic features (glucose tolerance test, area under the curve (GTT AUC), epididymal adipose tissue (EWAT) weight, fat gain, body weight gain and liver triglyceride levels). **b** N-integration discriminant analysis with DIABLO between cecal microbiota zOTUs and hepatic gene expression as determined by microarray analysis. $n = 10$ except for F where $n = 9$.

(Supplementary Fig. 5b–g). Non-significant changes in the same directions were also observed for mice fed diet A, indicating that the dietary lipids in diets A and C provoked similar shifts in both microbial composition and fermentation processes.

To investigate how dietary fatty acids affect the association between the gut microbiota and host features, we performed multivariate DIABLO modeling. We found that samples from diets A and C separated from other samples when integrating bacterial zOTUs with either host metabolic phenotype (Fig. 4a) or hepatic gene expression (Fig. 4b), indicating that these diets were the most important contributors of the observed coordinated changes in host and microbiota features. Notably, we found that abundance of zOTUs belonging to the *Roseburia* genus was negatively correlated to expression of hepatic genes associated with cholesterol synthesis (23-fold enrichment ($p = 0.0006$) of genes in GO:0006694, steroid biosynthetic process).

Taken together, we find that the diets used in the study, in particular diets A and C and to some extent diet B, affect gut microbiota diversity composition and function compared to MF diet, and that diets A and C drive correlated variation of bacterial taxa and host features.

### Cecum levels of long-chain SFA correlate with microbial taxa

The experimental diets gave rise to within- and between-group variation of fatty acid levels in cecum and serum (Supplementary Fig. 6a, b). We used this variation to explore associations between fatty acids and cecum bacteria. To measure covariation, we performed multi-omics data integration by regularized canonical correlation analysis (rCCA). The relative number of correlations between cecum fatty acids and zOTUs was 10-fold higher than between serum fatty acids and zOTUs (Fig. 5a). Specifically, cecum levels of long-chain SFA (C18:0 to C24:0) were correlated, mostly negatively, to the abundance of many zOTUs (Supplementary Fig. 6c and Supplementary Data 1). Samples from mice fed diets A and C were the main drivers of the concomitant variation between cecum fatty acids and bacteria (Fig. 5b). Therefore, elevated

levels of long-chain SFA might explain the decreased phylogenetic diversity and species richness observed for mice fed diets A and C (Fig. 3c and Supplementary Fig. 5a).

To gain better insight into the relationship between cecum SFA levels, microbiota structure and metabolic parameters, we performed network analysis. We observed that cecum levels of long-chain SFA were negatively associated with observed species diversity and proportions of *Roseburia, Anaerotruncus* and *Intestinimonas* and positively associated with proportions of *Akkermansia* and *Bacteroides* as well as *Christensenellaceae* R7 group (Fig. 5c and Supplementary Data 1). Of note, *Anaerotruncus* was in turn positively associated with impaired glucose tolerance, while *Akkermansia* and *Christensenellaceae* R7 group were negatively associated with adiposity and positively associated to each other (Fig. 5c and Supplementary Data 1).

rCCA was performed to also study covariation between fatty acids in cecum and serum and hepatic gene expression. Here the relative number of correlations was 10-fold higher for serum fatty acids than for cecum fatty acids (Supplementary Fig. 6d). The fatty acids covarying with hepatic genes clustered into two groups, one mainly including *n-3* PUFA and the other *n-6* PUFA and MUFA (Supplementary Fig. 6e). Both groups correlated with genes enriched in processes related to lipid metabolism (Supplementary Table 4). Samples from mice fed diets E and G, and to a lesser extent diets A and C, were the main drivers of the concomitant variation between serum fatty acids and hepatic gene expression (Supplementary Fig. 6f).

Taken together, we find that increased cecum levels of long-chain SFA are associated with a compositional and possible functional shift in the gut microbiota, while fatty acids in serum correlate with hepatic gene expression.

### Gut microbiota from mice fed diet A counteracts steatosis

The experiments described above show that mice fed diets A and C had improved metabolic profile and significant shifts in cecum microbiota composition and fermentation profiles compared to mice fed MF diet. These diets were also the main contributors to the across-diet correlation of host and microbial features and are therefore most likely to affect host metabolism through gut microbiota-dependent mechanisms. Given that mice fed diet A had lower hepatic steatosis (Fig. 2g) and better glucose tolerance (Supplementary Fig. 2k, l) than mice fed diet C, we selected diet A for further experiments to test the direct link between fat-induced microbiota alteration and metabolic phenotype.

First, we fed diet A or MF to germ-free mice for 9 weeks to determine if a gut microbiota is necessary for the development of steatosis. The dietary groups did not differ in weight gain and did not develop steatosis (Supplementary Fig. 7), demonstrating that the gut microbiota is necessary to provoke the metabolic perturbations induced by the MF diet.

Next, we tested whether the gut microbiota of mice fed diet A could attenuate the metabolic perturbations induced by the MF diet. In this experiment, after antibiotic treatment, mice fed MF diet were given a cecal microbiota obtained from mice fed either MF or diet A twice a week for 9 weeks (15 recipient mice per group). In line with the metabolic phenotypes observed for diet A, and with our hypothesis that those phenotypes were partly directly mediated by the gut microbiota, we observed that mice receiving diet A microbiota gained less weight and had improved glucose tolerance and less steatosis than mice receiving MF microbiota (Fig. 6a–d and Supplementary Fig. 8a–k). There was no difference for fecal levels of cholesterol, cholesteryl esters and triglyceride (Supplementary Fig. 8l–n). Consistent with the results displayed in Fig. 2h, mice receiving diet A microbiota had a trend towards higher hepatic expression of *Hmgcr* compared to mice receiving MF microbiota (Fig. 6e). Expression of genes encoding zonulin and occludin in ileum (Supplementary Fig. 9)

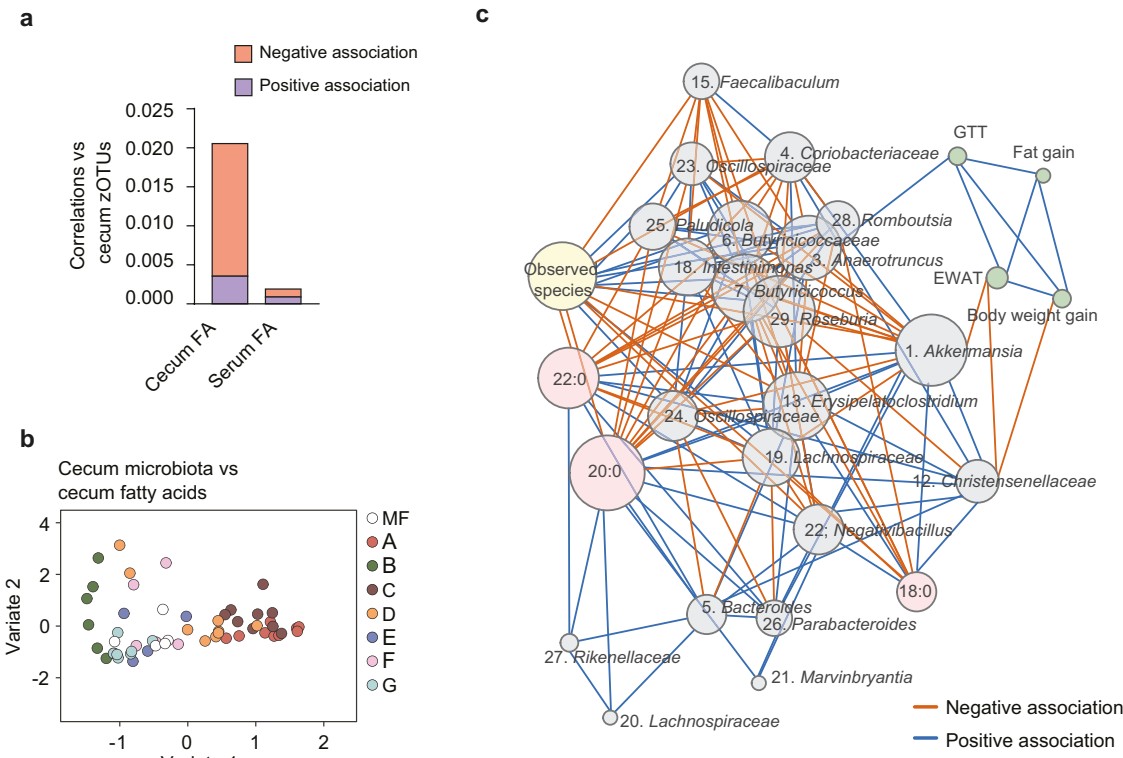

**Fig. 5 | Covariance between fatty acids, cecum microbiota and host features.** **a** Density of the correlation network between cecal zOTUs and cecal or serum fatty acids determined by regularized canonical correlation analysis (rCCA). **b** Cluster plot based on correlations between cecal fatty acids and cecal zOTUs determined by rCCA. **c** Network of interactions between cecum fatty acids, bacterial taxa, microbiota observed species and host metabolic phenotypes. Spearman's rho and taxonomy of bacterial taxa in Supplementary Data 1. $n = 5$ (MF), 10 (A), 6 (B), 9 (C), 7 (D), 4 (E), 3 (F), 7 (G). GTT glucose tolerance test, EWAT epididymal white adipose tissue. See also Supplementary Fig. 6 and Supplementary Data 1 and 2. Source data are provided as a Source data file.

and serum cytokine levels (Supplementary Table 5) did not differ between the groups. However, in line with the results shown in Fig. 2d and Supplementary Fig. 2g, mice receiving diet A microbiota had non-significant increased fecal levels of total free fatty acids (Fig. 6f) and strongly increased ratios of saturated vs unsaturated free fatty acids compared to mice receiving MF microbiota (Fig. 6g).

We performed 16S rRNA gene sequencing and microbial cell count by qPCR in cecum samples at the end of the treatment to determine the effects of microbiota administration in recipient mice fed MF diet. There was no difference in bacterial density between the groups (Fig. 7a). The number of observed species and phylogenetic diversity were lower in mice that received diet A microbiota than in mice that received MF microbiota, consistent with the differences in α-diversity seen in the inocula (Fig. 7b and Supplementary Fig. 10a). We also observed a significant difference for the cecal microbiota composition in mice given diet A microbiota compared to mice given MF diet microbiota (ANOSIM; Bray–Curtis: $p = 0.0001$, $R = 0.51$; unweighted UniFrac: $p = 0.0001$, $R = 0.81$; Fig. 7c, d). Interestingly, the ratio between fecal saturated and unsaturated fatty acids (Fig. 6g) in the recipient mice was negatively correlated with body weight ($R = −0.50$, $p = 0.02$, Supplementary Fig. 10b). This ratio also contributed to 15% of the compositional variation of the cecal microbiota and correlated with the relative abundance of the genera *Lactobacillus* and *Acetatifactor* (Supplementary Fig. 10c). Abundances of these bacteria were significantly increased in the recipients of diet A microbiota (Fig. 7e). Genera decreased in recipients of diet A microbiota included *Anaerotruncus*, which abundance was also associated with poor glucose tolerance in our network analysis (Fig. 5c), and *Akkermansia*, that was decreased even though the levels were increased when mice were fed diet A (Fig. 3f). Despite the changes in microbiota composition, we

found no difference for SCFA levels in the cecum (Supplementary Fig. 10d–i).

These results suggest that some of the observed changes in microbiota composition and metabolism, such as the increase in *Akkermansia* (Fig. 3f) and shift in SCFA profile induced by diet A (Supplementary Fig. 3b–g) were dependent on the interaction between the gut microbiota and dietary lipids. However, our results also indicate that the microbiota selected by diet A protects against adiposity, glucose intolerance and liver steatosis induced by the MF diet. Modulation of hepatic cholesterol synthesis and biohydrogenation of fatty acids in feces, which is a known property of *Lactobacillus* strains[17] might be mechanisms contributing to the beneficial effects of diet A microbiota.

## Dietary fatty acids affect *vena porta* bile acid levels

Bile acids are key mediators of microbiome-host interaction[18]. To identify metabolites linking the gut microbiota to the observed metabolic phenotypes we measured *vena porta* bile acids in mice fed diet A and MF. We found that mice fed diet A had higher total bile acid levels and tauro-beta-muricholic acid (TβMCA) levels than mice fed MF diet (Fig. 8a, b). Multivariate analysis of variance (adonis, 9999 permutations) showed that overall bile acid composition differed between mice fed diet A or MF ($p = 0.02$; Fig. 8c) and that the difference was driven by the proportions of TβMCA, tauro-omega-muricholic acid (TωMCA), the unconjugated forms of these bile acids βMCA and ωMCA as well as tauro-cholic acid (TCA) and cholic acid (CA). The ratio between TβMCA and its product of 6β-epimerization TωMCA[19] as well as the ratio between the unconjugated forms of these bile acids (βMCA/ωMCA) were higher in mice fed diet A (Fig. 8d). TβMCA is a potent antagonist of FXR[20] and we found that the expression of the

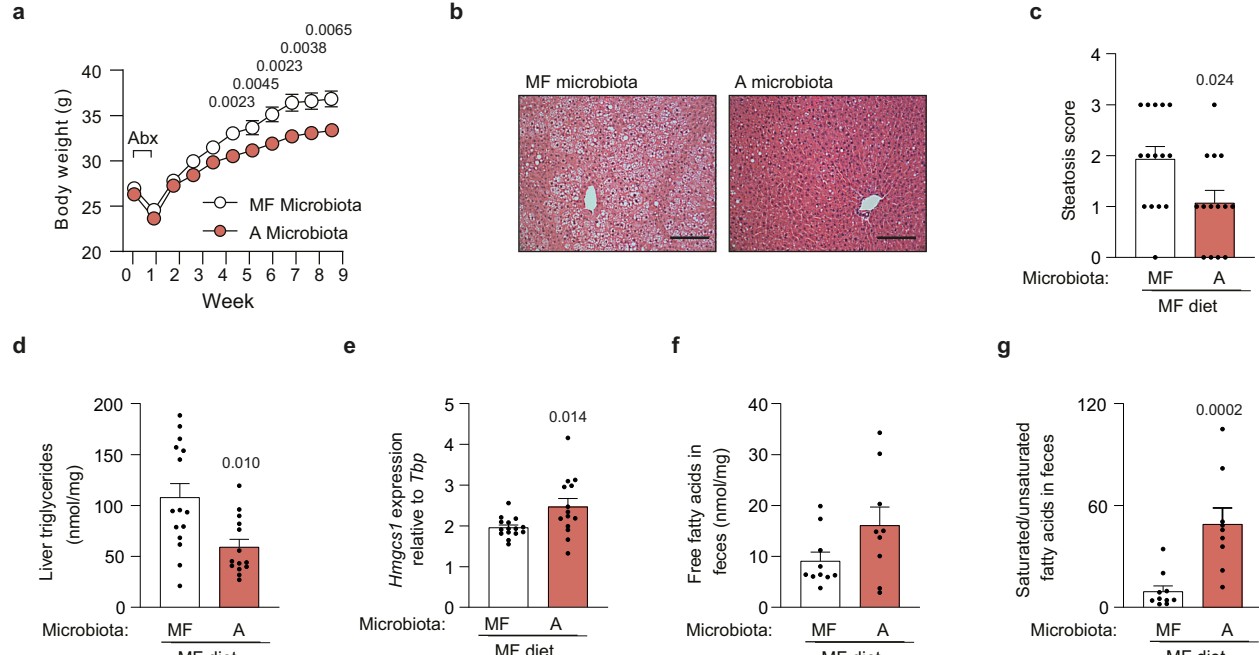

**Fig. 6 | Transfer of cecal microbiota from mice fed diet A protects against obesity, impairment of glucose metabolism and steatosis in mice fed the MF diet.** Mice were inoculated with cecal microbiota from mice fed MF or A diet and subsequently fed MF diet for 9 weeks. **a** Body weight, **b** eosin-hematoxylin staining of liver tissue (scale bar = 100 μm), **c** steatosis score, **d** liver triglyceride concentration, **e** hepatic expression of *Hmgcr* determined by qPCR, **f** fecal free fatty acids, and **g** ratio between saturated and unsaturated fecal free fatty acids. **a**–**e**: $n = 15$ (MF microbiota), 14 (A microbiota); **f**, **g**: $n = 10$ (MF microbiota) and 9 (A microbiota). Significant *p* values determined by two-sided Mann–Whitney *U*-test are displayed in the figure. Data are presented as mean ± SEM. *Hmgcs1* 3-hydroxy-3-methylglutaryl-Coenzyme A synthase 1. See also Supplementary Figs. 7 and 8. Source data are provided as a Source data file.

FXR targets *Fgf15* in the ileum (Fig. 8e) and *Shp* in the liver (Fig. 8f) were decreased in mice fed diet A.

To investigate if the differences in *vena porta* bile acid levels between mice fed diet A and MF were mediated by the gut microbiota, we measured bile acids in the mice transplanted with microbiota from mice fed MF or diet A. Total bile acid levels did not differ between the groups (Fig. 8g), but similar to mice fed diet A mice receiving diet A microbiota showed a non-significant increase in TβMCA levels (Fig. 8h), a shift in overall bile acid composition ($p = 0.007$; Fig. 8i) and significantly higher TβMCA/TωMCA and βMCA/ωMCA ratios compared to mice receiving MF microbiota (Fig. 8j). The expression of *Fgf15* in the ileum did not differ between the groups (Fig. 8k) but hepatic *Shp* expression was decreased in mice transplanted with diet A microbiota (Fig. 8l).

Our results show that gut microbiota from diet A directly modulates *vena porta* bile acid profiles and the expression of genes responsive to bile acids.

## Discussion

In the present study, we demonstrate that the amount of dietary SFA influences liver status and gut microbiota profile in humans. We also show in mice that variations in dietary fatty acid composition affect liver steatosis and metabolism and alter the gut microbiota. Compared to mice fed MF, mice fed diets enriched in PUFA had decreased steatosis and decreased lipogenesis, while mice fed diets enriched in stearic acid had decreased steatosis and increased cholesterol biosynthesis. Mice fed diets enriched in stearic acid also had increased cecum levels of long-chain SFA that correlated positively with the abundance of microbial taxa linked to improved host metabolism. The microbiota of mice fed diets enriched in stearic acid promoted increased saturation of fecal free fatty acid, gave rise to a shift in vena porta bile acid composition and protected against MF diet-induced steatosis and impaired metabolism.

In humans, we found that variations in SFA intake affected the gut microbiota more than variations in MUFA or PUFA intake. In line with our findings, a systematic review has shown that high SFA intake leads to decreased diversity and richness of the human gut microbiota, while PUFA diets have minor effects and reports on MUFA diets are inconsistent[21]. We controlled our analysis for intake of dietary fibers, which are known to have major impact on the gut microbiota[22]. Hence, despite a relatively low number of participants our results highlight the role of dietary lipids as an independent dietary factor shaping the bacterial community in the gut. In accordance with previous reports[7] we fund that impaired liver status was associated with decreased gut microbiota diversity. We also found that increased consumption of SFA and PUFA, but not MUFA, correlated with liver fat in individuals with obesity. Dietary intervention studies have shown that SFA raises liver fat levels more than PUFA and MUFA[23] while population-based studies provide limited data on the dietary risk factors associated with steatosis[24]. The association between dietary lipid composition and both gut microbiota structure and hepatic fat status indicates a potential role of the microbiota in mediating steatosis caused by dietary lipids.

To determine how variation in dietary fat affects associations between the gut microbiota, steatosis and host metabolism, we fed mice with diets that differed in lipid composition and used a steatosis-inducing MF diet as a reference. Consistent with earlier studies[25–28] we found that diets enriched in *n-3* PUFA (diets E and G) or medium-chain SFA (diet B) strongly reduced steatosis, while diets enriched in MUFA (diet D) or *n-6* PUFA (diet F) moderately reduced steatosis compared to MF diet. These diets had minor influence on the gut microbiota. In vitro experiments have shown that *n-3* PUFA regulates the activity of several transcription factors and decreases fatty acid synthesis in hepatocytes[29,30], and in agreement we showed that serum levels of *n-3* PUFA covaried with hepatic gene expression. We have previously shown that gut microbiota contributes to phenotypic differences in

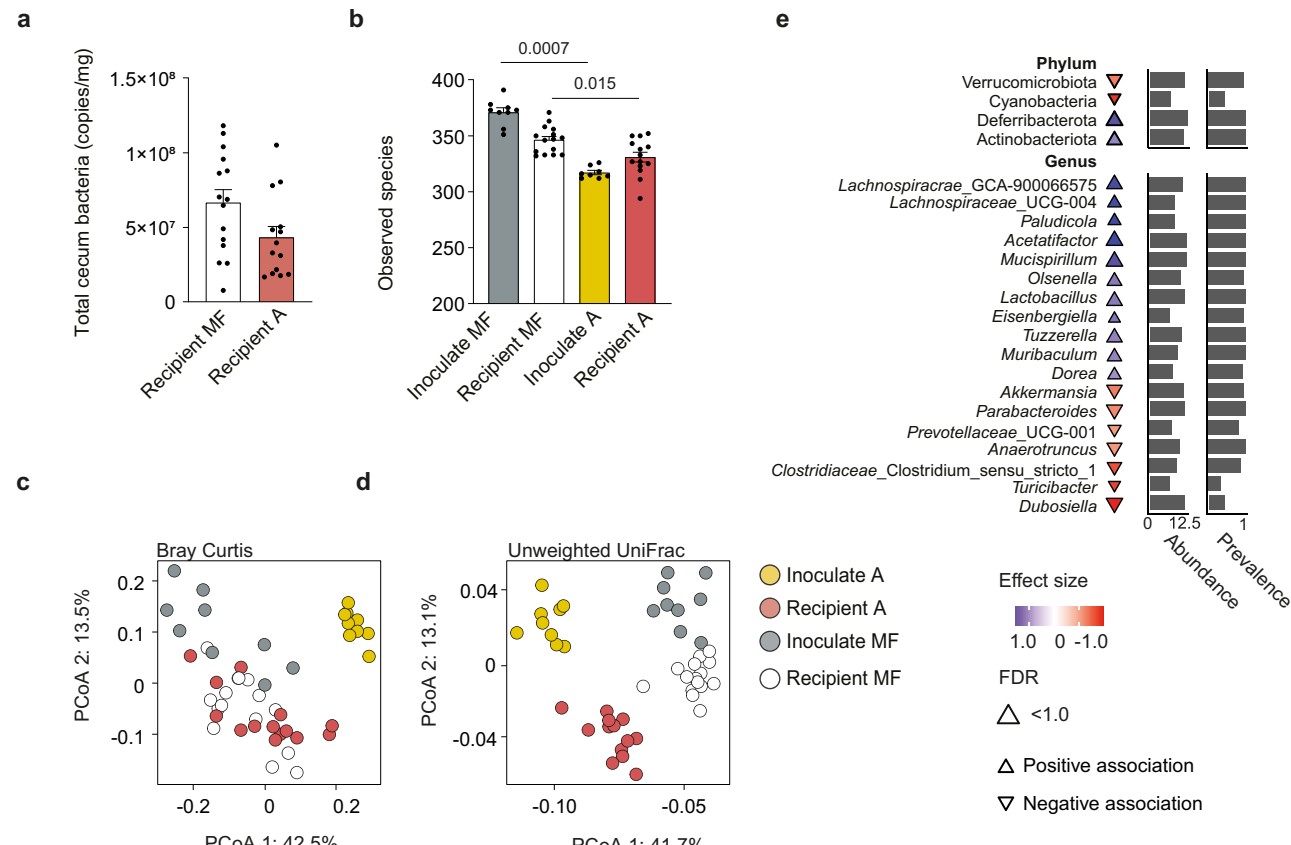

**Fig. 7 | Cecal microbiota composition in recipient mice after microbiota transfer.** Mice were inoculated with cecal microbiota from mice fed MF or A diet and subsequently fed MF diet for 9 weeks. **a** Quantification of total bacterial counts in cecum by qPCR. **b** Number of observed species of inoculums and in recipient mice fed MF microbiota and inoculated with either MF or A diet microbiota. Principal coordinates analysis (PCoA) based on **c** Bray–Curtis dissimilarity and **d** unweighted UniFrac distance (means of ranked distances between recipients of microbiota from mice fed diets A or MF and of the inoculums used to colonize the mice). **e** Differential abundance of bacterial taxa at phylum and genus levels for mice after inoculation with cecal microbiota from mice on MF or A diets. Two-sided Wilcoxon rank sum test FDR-adjusted $p$ value < 0.1. Direction and coloring of triangles indicate Cliff's Delta effect size (positive effect size and blue color indicate increased relative abundance in diet A microbiota recipients). The triangle size and left marginal bar plot depict log transformed abundance of individual taxa. The right marginal bar plot shows proportion of samples containing respective taxa. $n = 9$ (MF microbiota inoculum), 15 (MF microbiota recipients), 8 (A microbiota inoculum), 14 (A microbiota recipients). Significant $p$ values determined by two-sided Mann–Whitney $U$-test are displayed in the figure. Data are presented as mean ± SEM. See also Supplementary Fig. 10. Source data are provided as a Source data file.

mice fed lard, rich in saturated fat, or fish oil, rich in *n-3* PUFA, as sole fat source[11]. However, the present study indicates that moderate levels of dietary *n-3* PUFA mainly influence steatosis and hepatic metabolism through microbiota-independent mechanisms.

Our mouse studies showed that diets rich in stearic acid (diets A and C) reduced hepatic steatosis and obesity, and improved glucose tolerance compared to the MF diet. In contrast to other long-chain SFA, stearic acid has been shown to promote a healthy metabolic phenotype including decreased plasma cholesterol and reduced fat accumulation[31,32]. Stearic acid was also recently shown to prevent liver damage and to induce a shift in gut microbiota profile with increased *Akkermansia muciniphila* in a mouse model of alcoholic liver disease[33]. We found that diets A and C increased *Akkermansia* and had a major influence on microbiota communities, affecting composition, diversity and activity as indicated by the fermentation profiles. Stearic acid in diets A and C is mainly derived from cocoa fat where it is mainly esterified to triglycerides in position *sn-1* or *sn-3*. Stearic acid in these positions is poorly absorbed[34]. Cocoa fat does not induce metabolic perturbations in rabbits[35], or humans[36], potentially because of its low digestibility[37]. We demonstrated that mice fed diets A and C had reduced lipid absorption as shown by increased cecum levels of SFA, which were paralleled by higher abundance of bacterial taxa linked to improved host metabolic features. We also showed that transfer of the

microbiota from mice fed diet A into mice that were fed MF diet protected against obesity, impaired glucose homeostasis and liver steatosis. The ratios between saturated and unsaturated fecal fatty acids were increased in mice that were given diet A microbiota when fed a MF diet. This may be explained by an increased capacity of microbial biohydrogenation. Several strains in the genus *Lactobacillus*, which was increased in recipients of diet A microbiota, can saturate PUFA[17]. Metabolic intermediates in this process are known to promote metabolic health[38,39]. Finally, we found that differences in *vena porta* bile acid in mice fed diet A and MF were largely transmitted through gut microbiota transfer. Bile acids influence host metabolism via bile acid receptors such as farnesoid X receptor (FXR) and G-protein-coupled receptor-1 (TGR5) that regulate lipid, glucose, steroid, and energy metabolism[18]. Thus, even though other mechanisms such as metabolic endotoxemia may be involved, bile acids have the potential to mediate the microbiota-dependent difference in phenotype between mice fed diet A and MF. The increased ratio between substrates and products of 6β-epimerization in mice fed diet A indicates that bacteria able to perform this transformation are reduced by diet A.

To recapitulate, we provide insight into how defined changes in dietary fatty acid composition, independent of dietary fiber, influence gut microbiota profile, bile acid profile, host metabolism and liver

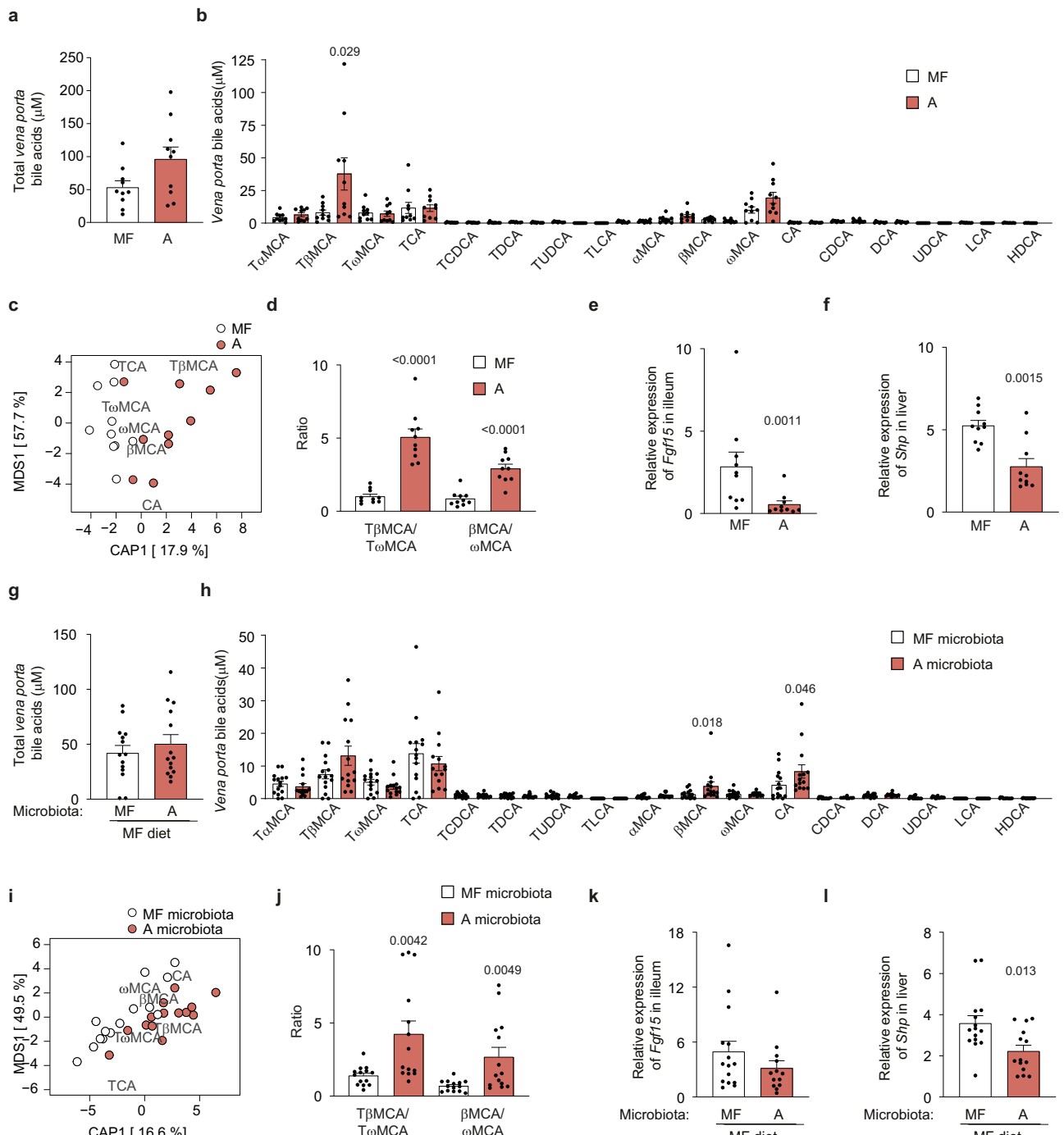

**Fig. 8 | The cecal microbiota from mice fed diet A regulates *vena porta* bile acid levels and hepatic expression of *Shp* in mice fed the MF diet.** Bile acid levels and expression of genes regulated by bile acids in mice fed diet A or MF diet for 9 weeks (**a**–**f**) and in mice inoculated with cecal microbiota from mice fed MF or A diet and subsequently fed MF diet for 9 weeks (**g**–**l**). **a** Total *vena porta* bile acid levels, **b** *vena porta* levels of individual bile acids, **c** redundancy analysis (RDA) plot based on relative bile acid levels in *vena porta*, **d** TβMCA/TωMCA and βMCA/ωMCA ratios, **e** relative ileum expression of *Fgf15* and **f** relative hepatic expression of *Shp* in mice fed diet A or MF diet. **g** Total *vena porta* bile acid levels, **h** *vena porta* levels of individual bile acids, **i** redundancy analysis (RDA) plot based on relative bile acid

levels in *vena porta*, **j** TβMCA/TωMCA and βMCA/ωMCA ratios, **k** relative ileum expression of *Fgf15*, and **l** relative liver expression of *Shp* in mice inoculated with cecal microbiota from mice fed MF or A diet and subsequently fed MF diet. **a**–**f**: $n = 10$; **g**–**l**: $n = 15$ (MF microbiota), 14 (A microbiota) except for **k** where $n = 13$ for A microbiota. Significant p-values determined by two-sided Mann–Whitney *U*-test are displayed in the figure. Data are presented as mean ± SEM. MCA muricholic acid, CA cholic acid, CDCA chenodeoxycholic acid, DCA deoxycholic acid, UDCA ursodeoxycholic acid, LCA lithocholic acid, HDCA hyodeoxycholic acid, T taurine-conjugated species, *Fgf15* fibroblast growth factor 15, *Shp* small heterodimer partner. Source data are provided as a Source data file.

steatosis. The presented data show that interaction between the gut microbiota and dietary lipids can be used to modulate liver steatosis. Furthermore, it suggests that fatty acid absorption is a key factor determining dietary influence on the microbiota in distal parts of the gut. We find that dietary fatty acid composition also affects gut microbiota profile in a human cohort, supporting clinical relevance. The interplay between dietary fatty acids and gut microbiota can be harnessed to improve dietary recommendations for metabolic health.

## Methods

### Ethics

The research performed in this study complies with all relevant ethical regulations. The Institutional Review Board - Ethics Committee and the Committee for Clinical Research of Dr. Josep Trueta University Hospital (Girona, Spain) approved the study protocol of the human study, and the University of Gothenburg Animal Studies Committee approved the study protocol of the mouse study.

### Human subjects

Subjects from the human cohort ($n = 117$; 79 women, 35 men) were recruited at the Endocrinology Department of Dr. Josep Trueta University Hospital, Girona, Spain. The recruitment of subjects started in January 2016 and finished in October 2017. Consecutive middle-aged subjects, 27.2–66.6 years, were included. Subjects with obesity (body mass index (BMI) $\geq 30$ kg/m$^2$, $n = 65$) and age-matched and sex-matched subjects without obesity (BMI 18.5–30 kg/m$^2$, $n = 52$) were eligible. Exclusion criteria were type 2 diabetes mellitus, chronic inflammatory systemic diseases, acute or chronic infections in the previous month; use of antibiotic, antifungal, antiviral or treatment with proton pump inhibitors; severe disorders of eating behavior or major psychiatric antecedents; neurological diseases, history of trauma or injured brain, language disorders and excessive alcohol intake ($\geq 40$ g/day in women or 80 g/day in men). Population characteristics has been previously described[40]. Informed written consent was obtained from all participants and participants received no compensation. Stools were collected within 4 h after their emission and preserved at −80 °C.

Dietary characteristics of the subjects were collected in a personal interview using a validated semi-quantitative food-frequency questionnaire (FFQ), including 93 different kinds of foods and beverages commonly consumed in Spain (available at: http://epinut.edu.umh.es/en/cfa-93-encv/). The FFQ is a Spanish version of the Harvard Questionnaire[41] which was modified and validated for the Mediterranean area of Spain[42]. Participants were asked how often on average they had consumed each food item during the previous year. Serving sizes were specified for each food item in the FFQ. The FFQ had nine possible responses, ranging from "never or <1 per month" to "6 or more per day." Average daily nutrient intakes were obtained by multiplying the consumption frequency of each food item by the nutrient content in the specified portion/serving specified on the FFQ. The nutrient values were obtained from the food composition tables of the US Department of Agriculture (http://www.nal.usda.gov/fnic/foodcomp), which includes data on practically all food items available in Spain and on a wide variety of ethnic foods, as well as other published sources for Spanish foods and portion sizes.

To investigate how consumption of dietary fats with different fatty acid compositions affects gut microbiota composition in humans, we divided subjects ($n = 117$, $n = 52$ lean, $n = 65$ obese) into tertiles (low, medium, and high consumers) based on their consumption of saturated fatty acids (SFA, $n = 39$ per tertile), monounsaturated fatty acids (MUFA, $n = 39$ per tertile) or polyunsaturated fatty acids (PUFA, $n = 39$ per tertile).

Study outcomes were liver fat determined by MRI and FLI, consumption of saturated, monounsaturated, and polyunsaturated fatty acids and gut microbiota features including taxa abundance and measures of α-diversity. Potential confounder included age, body mass index, sex and intake of dietary fibers.

### Liver fat determined by MRI and calculation of FLI

A 1.5 Tesla scanner (Ingenia Philips Medical Systems, Eindhoven, The Netherlands) with an 8-channel receiver-coil array was used to examine all participants. The imaging protocol comprised a three-dimensional volumetric mDixon gradient-recalled echo acquisition in the axial plane that covered the entire abdominal region. We reconstructed images of fat, water, in-phase and out-of-phase (field-of-view 230 × 190 mm,

repetition time 5.9 ms, excitation time 1.8 and 4 ms, flip angle 15°, number of slices 50, voxel size 1 × 1 × 1 mm, thickness 10 mm, acquisition time 6 min). The first array was acquired from the diaphragm to the kidneys, and the second from the kidneys to below the pubic symphysis. Each stack was acquired in 11 s breath-hold task. Images of fat and water were input into Olea Sphere 3.0 to calculate the fat fraction map. Two regions of interest (40 pixels) were manually performed to determine the mean value of the right lobe of the liver, and the average was then calculated. The regions of interest were examined while avoiding breathing-related or susceptibility-related motion artifacts.

FLI, an algorithm based on waist circumference, body mass index (BMI), triglyceride, and gamma-glutamyl-transferase (GGT) for the prediction of fatty live, was calculated as previously been described[43]

### Extraction of fecal DNA and shotgun sequencing in humans

Total DNA was extracted from frozen human stools using the QIAamp DNA mini stool kit (Qiagen, Courtaboeuf, France). Quantification of DNA was performed with a Qubit 3.0 fluorometer (Thermo Fisher Scientific, Carlsbad, CA, USA), and 1 ng of each sample (0.2 ng/µl) was used for shot gun library preparation for high-throughput sequencing, using the Nextera DNA Flex Library Prep kit (Illumina, Inc., San Diego, CA, USA) according to the manufacturers' protocol.

Sequencing was carried out on a NextSeq 500 sequencing system (Illumina) with 2 ×150-bp paired-end chemistry, at the facilities of the Sequencing and Bioinformatic Service of the FISABIO (Valencia, Spain). The obtained input fastq files were decompressed, filtered and 3· ends-trimmed by quality, using prinseq-lite-0.20.4 program[44] and over-lapping pairs were joined using FLASH-1.2.11[45]. Fastq files were then converted into fasta files, and host reads were removed by mapping the reads against the GRCh38.p11, reference human genome (Dec 2013), by using bowtie 2-2.3.4.3[46] with end-to-end and very sensitive options. Taxonomic annotation, was implemented with Kaiju v1.6.2 on the filtered reads[47]. Addition of lineage information, counting of taxa and generation of an abundance matrix for all samples were performed using the package R (v.3.1.0).

### Analysis of metagenomics data from human feces samples

Zero counts in the shotgun metagenomics data were imputed using a geometric Bayesian-multiplicative replacement with the zCompositions (v.1.4.0-1) R package[48] and the "cmultRepl" function. To consider the compositional structure of the microbiome, counts were then transformed using a centered log-ratio (clr) transformation as implemented in the "compositions" R package[49]. Bacterial species associated with the dietary fatty acids were then identified using robust linear regression models as implemented in the Limma (v.3.30.13) R package[50], adjusting for age, BMI, sex and fiber consumption. Taxa were previously filtered so that only those with more than 10 reads in at least two samples were selected. The $p$ values were adjusted for multiple comparisons using a Sequential Goodness of Fit[51] as implemented in the SGoF (v2.3.2) R package. Statistical significance was set at $p_{adj} < 0.1$.

### Mouse experiments

C57Bl/6 mice were maintained at a temperature of $20 \pm 1$ °C, an air humidity of 45–70% at a 12-h light/dark cycle (light from 7 a.m. to 7 p.m.) under standard specific-pathogen-free (SPF) or GF conditions. SPF mice were purchased from Taconic while GF mice were bred in-house. All mice were males, 8 weeks old and weight matched at the start of the experiments. Mice were fed irradiated isocaloric diets differing only in their composition of fat (SPF mice: 10 mice per group, 5 mice per cage, 2 cages per group, in total 80 mice (1 mouse died); GF mice: 9 or 7 mice per group, 2–5 mice per cage, 2 cages per group, in total 16 mice; Envigo TD.180342, TD.180343, TD.180344, TD.180345, TD.180346, TD.180347, TD.180348, TD.180349; 42% kcal fat, 0.2% weight total cholesterol; Fig. 2a, b), for 9 weeks. The mice were fasted for 4 h before they were killed. Tissue, serum and intestinal content

samples were harvested at the end of the experiment. Weekly food consumption was measured cage-wise.

Gut microbiota administration with cecal content from donor mice (15 donors per group, 5 mice per cage, 3 cages per group) was performed on 8-week-old male mice (15 mice per group, 5 mice per cage, 3 cages per group, in total 30 mice (1 mouse died) kept on chow diet. Before the administration, recipient mice were treated with an antibiotic cocktail (neomycin 100 µg/ml, streptomycin 50 µg/ml, ampicillin 100 µg/ml, vancomycin 50 µg/ml, metronidazole 100 µg/ml, bacitracin 1000 µg/ml, ciprofloxacin 125 µg/ml and ceftazidime 100 µg/ml) given in the drinking water ad libitum for 3 days. During the last 12 h of antibiotic treatment, the mice were fed with the diets of the donor mice to facilitate subsequent colonization. Antibiotic treatment was stopped 12 h before the first microbiota administration. Gavage with cecal microbiota was performed twice every week throughout the whole feeding period. Mice were colonized twice a week by oral gavage with 200 µL of cecal suspension after a 4-h fast. Gavage solutions were prepared by dissolving cecal content from three donor mice (fed either MF or diet A) in 10 mL of LYBHI medium (brain-heart infusion medium supplemented with 0.5% yeast extract, 1 mg/ml cellobiose, 1 mg/ml maltose, and 0.5 mg/ml cysteine) in an oxygen-free environment and were used fresh (first gavage each week) or frozen (second gavage each week). After the first microbiota gavage, all mice were fed a MF diet for 9 weeks. The last gavage was performed 3 days before the mice were killed. A priori exclusion criteria were loss of >10% body weight and/or worsened general health condition.

## Analysis of lipids in mice

Total fatty acids in mouse diet, cecum and serum from *vena cava* were quantified by fatty acid methyl ester (FAME) analysis. 950 µl toluene, 50 µl 1000 ppm of $C_{23}$ stock solution and 1 ml of freshly prepared methanolic HCl (10% v/v) were added. Samples were incubated at 70 °C for 120 min. 1.0 ml Milli-Q water and 1.0 ml hexane were added, and samples were vortexed for 60 s and centrifuged at $100 \times g$ for 6 min. The upper phase was transferred to clear GC vials. FAMEs were analyzed using a Thermo ISQ-LT GC-MS system. Separation of FAMEs was performed on a Zebron ZB-FAME GC column (20 m × 0.18 mm I.D., 0.15-µm film thickness) (Phenomenex, Macclesfield, UK). An external FAME standard mix GLC 403 (Nu-Chek Prep, Inc., Elysian, MN, USA) was used for identification of peaks.

Liver and fecal lipids were extracted using the BUME[52] method and free cholesterol, triglycerides and cholesteryl esters were quantified using straight phase HPLC with evaporative light scattering detection. To analyze fecal free fatty acids, BUME samples were evaporated to dryness ($N_2$, 40 °C) and 100 µl methyl-tert-bythyl ether, 20 µl methanol and 50 µl trimethylsilyldiazomethane were added. Samples were mixed for 5 min, left at room temperature for 15 min, evaporated at room temperature under $N_2$ and dissolved in 150 µl ethyl acetate. Samples were quantified using gas chromatography (Agilent Technologies 7890A) with mass spectrometric detection (Agilent Technologies 5975C). Fatty acid methyl ester standards were obtained from Merck (Darmstadt, Germany).

All lipid analyses were preformed blinded. Some fecal lipids values are missing due to insufficient availability of biological material (Fig. 2 and Supplementary Fig. 2).

## Analysis of metabolic and inflammatory parameters in mice

Fat and lean mass were determined by magnetic resonance imaging. Glucose tolerance tests were performed by injecting glucose (2 g/kg body weight) intraperitoneally after a 5 h fast. Tail blood samples were collected at −30, 0, 15, 30, 60, 90, and 120 min and blood glucose levels were determined using a glucose meter (Accu Check Aviva, Roche). The glucose tolerance test for the diet experiment (data presented in Supplementary Fig. 2k, l) was performed over 4 days (20 mice per day) with the cages from each dietary group tested on different

days. In the glucose tolerance tests for the microbiota transfer experiment (data presented in Supplementary Fig. 7f, g) measurements were made on every other cage with MF microbiota recipients and every other cage diet A recipient. Insulin levels were measured with a Crystal Chem kit (Downers Grove, IL, USA) according to the manufacturers' protocols. Paraffin-embedded epididymal adipose tissue and liver sections (7 µm) were stained with hematoxylin-eosin and quantified by densitometric analysis using Biopix iQ software (v. 2.1.3; Biopix, Sweden). Hepatic steatosis and lobular inflammation score were semi-quantitatively determined by blinded histological evaluation[53]. Serum levels of cytokines in systemic circulation were determined using a V-PLEX Proinflammatory Panel 1 Mouse Kit (K15048D, Mesoscale) according to manufacturer's protocols.

## Microarray analysis of mouse liver samples

Total RNA was extracted with beta-mercaptoethanol using RNeasy kit (QIAGEN, Hilden, Germany). Gene expression profiles were obtained at the GeT-TRiX facility (GénoToul, Génopole Toulouse Midi-Pyrénées, France) using Sureprint G3 Mouse GE v2 microarrays (8 × 60 K; design; 074,809; Agilent Technologies, Santa Clara, CA, USA) following the manufacturer instructions. Microarray data were processed using Bioconductor (v.3.0)[54]. Raw data (median signal intensity) were filtered, log2 transformed and normalized using quantile method[55]. Microarray data and experimental details are available in NCBI's Gene Expression Omnibus and are accessible through GEO Series accession number GSE222060.

## Extraction of genomic DNA and 16S rRNA sequencing in mice

Total genomic DNA was isolated from 50–100 mg of cecum using repeated bead beating and the Nucleospin® Soil kit. Samples were placed in SL2 buffer with SX enhancer, incubated at 90 °C for 10 min and sheared with 6 rounds of bead beating at 5.5 m/s for 60 s in a FastPrep®-24 Instrument (MP Biomedicals). The V4 region of the 16 S rRNA gene was amplified with primers 515 F and 806 R in duplicate PCR reactions in 25 µl volumes containing 1× 5PRIME HotMasterMix (5PRIME), 200 nM of each primer, 0.4 mg/ml BSA, and 5% dimethylsulfoxide. The PCR program consisted in an initial denaturation for 3 min at 94 °C; followed by 25 cycles of denaturation for 45 s at 94 °C, annealing for 60 s at 52 °C and elongation for 90 s at 72 °C; and a final elongation step for 10 min at 72 °C. Duplicates were combined, purified with the NucleoSpin Gel and PCR Clean-Up kit (Macherey-Nagel), and quantified using the Quant-iT PicoGreen dsDNA kit (Invitrogen). Purified amplicons were diluted to 10 ng/µl, pooled in equal amounts and purified again using Ampure magnetic purification beads (Agencourt) to remove short products. Negative controls were run for each sample and the absence of detectable PCR products was confirmed with gel electrophoresis.

Amplicons were sequenced in a MiSeq instrument (RTA (v.1.17.28), bundled with MCS (v.2.5; Illumina) with the V2 kit (2 × 250 bp paired-end reads; Illumina). Illumina reads were merged using Usearch (v.11) 64-bit (Edgar, 2010) allowing for up to 30 mismatches in the alignment of the paired-end reads while discarding reads with a merged length greater than 270 bp and fewer than 230 bp. The merged reads were quality filtered based on expected errors removing reads above the threshold of 1[56]. The merged reads were turned into zero-radius operational taxonomic units (zOTUs) by compiling the sequences into sets of unique reads and performing error-correction using the UNOISE3 algorithm discarding sequences with fewer than 4 reads. The zOTUs were assigned taxonomy using DADA2's assignTaxonomy (minBoot = 50) and assignSpecies, using formatted version of the Silva v.138 database. A phylogenetic tree of the sequences was created with the help of the MAFFT software (v.7.407[57] and the FastTree software (v.2.1.10).

Before analysis the data were filtered to remove low abundant and highly sparse zOTUs. The zOTUs were filtered based on fraction of total abundance, retaining all zOTUs contributing to more than 0.002% of the total amount of reads. For the main study 99.7% of the

reads (6,588,109) were kept and 51% (518) of the original zOTUs. For the cecal microbiota transfer study 99.8% of the reads (8,310,323) were kept and 62% (428) of the original zOTUs.

## Quantitative RT-PCR

Quantitative PCR was used to enumerate total bacterial 16 S rRNA gene copies in the genomic DNA extracted from cecal samples as described previously[58]. Determination of the relative gene expression in the liver and ileum tissue was conducted by qPCR as described previously[59]. The primer sequences used in this study were *Tbp*-F: GAAGAACA ATCCAGACTAGCA, *Tbp*-R: CCTTATAGGGAACTTCACATCACA, *Hmgcr*-F: AGCTTGCCCGAATTGTATGTG, *Hmgcr*-R: TCTGTTGTGAACCATGT GACTTC, *Hp*-F: GCCGCTAAGAGCACAGCA, *Hp*-R: TGGAGGTTTCCC CACTCTGA, *Ocln*-F: GGTTGATCCCCAGGAGGCTA, *Ocln*-R: TAGTCAG ATGGGGGTGGAGG.

## Measurement of SCFA in mice

Mouse cecum SCFA were measured using gas chromatography coupled to mass spectrometry detection (GC-MS). Approximately 20–100 mg of cecal content was mixed with internal standards, added to glass vials, and freeze-dried. All samples were then acidified with HCl, and SCFA were extracted with two rounds of diethyl ether extraction. The organic supernatant was collected, the derivatization agent *N*-tert-butyldimethylsilyl-N-methyltrifluoroacetamide (Sigma-Aldrich) was added and samples were incubated at room temperature overnight. SCFA were quantified with a gas chromatograph (Agilent Technologies, 7890A) coupled to a mass spectrometer (Agilent Technologies, 5975C). SCFA standards were attained from Sigma-Aldrich (Stockholm, Sweden).

## Analysis of microarray data from mouse liver samples

The linear model was fitted using the limma lmFit function[50] considering washing batches as a blocking factor. Pairwise comparisons between specific diets and MF diet were applied using specific contrasts. A correction for multiple testing was applied using the Benjamini–Hochberg procedure for false discovery rate (FDR). Probes with FDR ≤ 0.05 and fold change >1.5 compared to MF diet were considered differentially expressed. GO analysis was performed and redundancy within lists of GO terms was reduced by the online software Revigo (v.1.8.1) with a similarity score set to 0.5. Analyses of transcription factor enrichment were performed with Enrichr[60] (v.2.1) by interrogating the TRRUST Transcription Factors 2019 database and reporting adjusted p values for mouse transcription factors while discarding human transcription factors.

Sparse partial least squares regression using the regression mode was used to select liver mRNAs that were most likely to predict liver triglyceride content. Variable selection is achieved by introducing LASSO penalization on the pair of loading vectors[61,62].

## Analysis of 16S rRNA data from mouse cecum samples

The graphical representations and statistical analyses of the microbiota were performed using R v.3.6.1 with packages phyloseq v.1.28[63] and ggplot2 (v.3.3.3)[64]. Faith's phylogentic diversity and richness were calculated using the Picante (v1.8.2-package)[65]. The analysis of β-diversity was performed on relative abundance normalized data. Bray–Curtis dissimilarity and unweighted UniFrac distance were calculated and principle coordinates analysis (PCoA) was performed with the vegan (v.2.5-6) R package[66]. The Adonis test with 9999 permutations was used to assess analysis of variance in of composition and was complemented by analysis of similarities ($n = 9999$) (ANOSIM) using the vegan package.

zOTUs were rarefied to read depth of the smallest sample ($n = 20,701$) using R-package rtk (v. 0.2.6.1) and binned on genus and phylum level according to taxonomic assignment using R-package phyloseq (v. 1.3.0). Differential abundance analyses were performed using R-package metadeconfoundR (v. 0.2.6) under standard settings:

Associations between rarefied bacterial read counts and diet were first tested using simple rank-based Wilcoxon tests. Using the "randomVar" parameter, we then controlled for cage effects in the detected associations by running an additional set of linear mixed-effects model likelihood ratio tests, adding the cage as random effect in all "diet X versus MF diet" comparisons (Figs. 3e, f and 7e).

## Multi-omics data integration

Bidirectional correlations between cecal OTUs and hepatic transcriptome or metabolic features of the host were investigated using N-integration discriminant analysis with DIABLO, an algorithm that aims to identify and select a highly correlated multiomics signature, while also maximizing discrimination between experimental groups[67] using the R package mixOmics (v6.10.9). Low abundance OTUs (representing <0.01% of total OTUs) were removed leading to a dataset of 291 OTUs, and the 5000 hepatic genes with the highest variance were kept. We used two components in the models, and for the estimation of model parameters, the cross-validation procedure (CV) method was used. For the correlation networks, only correlations with a Spearman's rank correlation coefficient > 0.75 were plotted.

Bidirectional correlations between fatty acids, measured in cecal content or in serum, and cecal OTUs or liver transcripts were investigated using the regularized canonical correlation analysis (rCCA) implemented in mixOmics. This algorithm maximizes the correlation between two matching omics datasets measured on matching samples. Models using cecal fatty acids were built using $n = 56$ mice for which cecal fatty acids were measured, while those using serum fatty acids were built with $n = 79$ mice. OTUs and liver mRNA were filtered as described above for DIABLO analysis. The density of the correlation network was calculated by dividing the number of correlated features with $\rho > 0.55$ by the total number of possible correlations in the model.

## Measurement of bile acids

Bile acids were analyzed using ultra-performance liquid chromatography-tandem mass spectrometry (UPLCMS/MS) according to previous work[68]. Briefly, samples were extracted with methanol containing deuterated internal standards. After 10 min of vortex and 10 min of centrifugation at $20,000 \times g$, the supernatant was diluted 100 times in methanol:water [1:1]. The bile acids were separated on a C18 column (1.7 μ, 2.1 × 100 mm; Kinetex, Phenomenex, USA) using water and acetonitrile as mobile phases and detected using MRM in negative mode on a QTRAP 5500 mass spectrometer (Sciex, Concord, Canada). Quantification was made using external standard curves.

## Statistics and reproducibility

Statistical analyses were performed in GraphPad Prism (version 9.1.2) unless otherwise stated. No data were excluded from the analyses. No statistical method was used to predetermine sample size. The experiments were not randomized. Analysis of lipids and evaluation of histology was performed blinded. For all other experiments the investigators were not blinded to allocation during experiments and outcome assessment.

## Reporting summary

Further information on research design is available in the Nature Portfolio Reporting Summary linked to this article.

# Data availability

The raw metagenomic sequence data of the 114 human subjects from the Ironmet cohort have been deposited in the European Nucleotide Archive (ENA) under the project number PRJEB39631 with the accession numbers ERS4859818–ERS4859933 (https://www.ebi.ac.uk/ena/browser/view/PRJEB39631). The microarray data generated in this study have been deposited in NCBI GEO under accession code GSE222060. The mouse cecum 16S rRNA data generated in this study

have been deposited in the European Nucleotide Archive (ENA) under accession code PRJEB58626. Data generated in this study are provided in the Source data file. The reference human genome GRCh38.p11 can be accessed here: https://www.ncbi.nlm.nih.gov/datasets/genome/GCF_000001405.37/. Silva can be accessed here: https://www.arb-silva.de/. Source data are provided with this paper.

## Code availability
No new code was created in this study.

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

## Acknowledgements

We thank Anna Hallén, Louise Helldén, Carina Arvidsson, Zakarias Gulic and Per-Olof Bergh for technical assistance, Rosie Perkins for commenting and editing the manuscript and Martin Adiels and Chinmay Dwibedi for help with mathematical calculations. We also thank Claire Naylies and Yannick Lippi from the GeT-TRiX facility at Toxalim for their contribution to microarray fingerprints acquisition and microarray data analysis carried out at GeT Genopole Toulouse Midi-Pyrénées facility. Measurements of fatty acids in diet, cecum and serum were performed at CMSI, Chalmers University. Computations were enabled by resources provided by the Swedish National Infrastructure for Computing (SNIC) at the Uppsala Multidisciplinary Center for Advanced Computational Science (UPPMAX) under project SNIC 2022/5–451 partially funded by the Swedish Research Council through grant agreement no. 2018-05973. This work was supported by the European Joint Programming Initiative "A Healthy Diet for a Healthy Life" (JPI HDHL) coordinated by Rémy Burcelin, the respective national/regional funding organizations: Svenska Forskningsrådet FORMAS (2017-02001). This work was further supported by the Deutsche Forschungsgemeinschaft (DFG, German Research Foundation) as part of the clinical research unit (CRU339); Food allergy and tolerance (FOOD@) —409525714. This work was also partially supported by research grants FIS (PI18/01022 and PI21/01361) and the project PI20/01090 (co-funded by the European Union under the European Regional Development Fund (FEDER) "A way to make Europe") from the Instituto de Salud Carlos III from Spain, by *Fondo Europeo de Desarrollo Regional* (FEDER) funds, and the Catalan Government (AGAUR, #SGR2017-734, ICREA Academia Award 2021), J.M.-P is funded by a Miguel Servet contract (CP18/00009) from the Instituto de Salud Carlos III.

## Author contributions

Conceptualization R.C.; investigation M.S., R.C., S.E.-S., J.D.K., M.M.-N., J.P., A.P.; formal analysis T.B., S.E.-S., J.M.-P., L.O., U.L., H.B. and M.S. with input from S.K.F.; writing—original Draft R.C.; writing—review and editing M.S., S.E.-S., T.B., J.M.-P., L.O., U.L., A. Moschetta, A. Montagner, P.G., C.H., H.G., V.T., J.M.F.-R., S.K.F., R.B.; funding Acquisition R.C., R.B. and S.E.-S.; project administration R.C. and R.B.

## Funding

## Competing interests

The authors declare no competing interests.

## Additional information

[1]The Wallenberg Laboratory, Department of Molecular and Clinical Medicine, Institute of Medicine, Sahlgrenska Academy, University of Gothenburg, 413 45 Gothenburg, Sweden. [2]Toxalim (Research Center in Food Toxicology), INRAE, ENVT, INP- PURPAN, UMR 1331, UPS, Université de Toulouse, Toulouse, France. [3]Max Delbrück Center for Molecular Medicine in the Helmholtz Association, 13125 Berlin, Germany. [4]Charité-Universitätsmedizin Berlin, Corporate Member of Freie Universität Berlin, Humboldt-Universität zu Berlin, and Berlin Institute of Health, 10117 Berlin, Germany. [5]Department of Diabetes, Endocrinology and Nutrition, Dr. Josep Trueta University Hospital, Girona, Spain. [6]Nutrition, Eumetabolism and Health Group, Girona Biomedical Research Institute (IdibGi), Girona, Spain. [7]CIBER Fisiopatología de la Obesidad y Nutrición (CIBERobn), Instituto de Salud Carlos III, Madrid, Spain. [8]Experimental and Clinical Research Center, A Cooperation of Charité-Universitätsmedizin Berlin and Max Delbrück Center for Molecular Medicine, Lindenberger Weg 80, 13125 Berlin, Germany. [9]DZHK (German Centre for Cardiovascular Research), Partner Site, Berlin, Germany. [10]Department of Radiology, Biomedical Research Institute Imaging Research Unit, Diagnostic Imaging Institute, Doctor Josep Trueta University Hospital of Girona, Avinguda de França, s/n, 17007 Girona, Catalonia, Spain. [11]Department of Interdisciplinary Medicine, University of Bari "Aldo Moro", Piazza Giulio Cesare 11, 70124 Bari, Italy. [12]Medicina e Chirurgia d'Accettazione E d'Urgenza, Azienda Ospedaliero-Universitaria Policlinico di Bari, 70124 Bari, Italy. [13]Medicina Sub-Intensiva, Presidio Maxi-Emergenze Fiera del Levante, Azienda Ospedaliero-Universitaria Policlinico di Bari, 70124 Bari, Italy. [14]Institut des Maladies Métaboliques et Cardiovasculaires, INSERM UMR 1297, Université Paul Sabatier, Université de Toulouse, F-31432 Toulouse, France. [15]Endocrinology-Diabetology-Nutrition Department, Toulouse University Hospital, Toulouse, France. [16]Department of Medical Sciences, Faculty of Medicine, Girona University, Girona, Spain. [17]Structural and Computational Biology Unit, European Molecular Biology Laboratory, 69117 Heidelberg, Germany. [18]These authors contributed equally: Marc Schoeler, Sandrine Ellero-Simatos, Till Birkner. ✉e-mail: robert@wlab.gu.se

