## [Peer Review File · Nature Communications]

REVIEWER COMMENTS

Reviewer #1 (Remarks to the Author):

This is a very interesting work describing the interaction between dietary lipids and gut microbiota in liver steatosis. Overall the preclinical studies are well performed and the results are consistent. I would suggest, however, that the authors include the analysis of gut permeability in the different models, given its relevance for NAFLD pathogenesis.

With regard to the clinical data, I have several comments. First, I would include in Figure 1 the PCA data that the authors mention in the text. Also, in my opinion the authors need to include an additional cohort of patients with characterized NAFLD (either liver biopsy or imaging methods) with FFQ and microbiota analysis. This sub-analysis will further confirm the hypothesis of the work.

Reviewer #2 (Remarks to the Author):

Schoeler et al. have provide insight into how changes in dietary fatty acid composition, independent of dietary fiber, influence gut microbiota profile, host metabolism and liver steatosis. This paper uses a translational approach using human data but rather focusses on obtaining mechanistic insight using rodent models. The paper holds clinical relevance and is of importance to the field. Nevertheless, the paper has not addressed how the gut microbial alterations induced by dietary fatty acids induce hepatic steatosis. I expect that some effects are mediated or driven by microbial metabolites. Adding data on circulating plasma metabolites (SCFA, bile acids or other microbial metabolites) would significantly improve the paper.

I have a few additional comments:

1. The paper starts by showing results from the IRONMET study (n=117) where the individuals were divided into tertiles based on SFA, MUFA or PUFA consumption. Despite showing that SFA decreases gut microbiota diversity and abundance of butyrate producing bacteria, the authors have not measured SCFA in the human cohort. Adding SCFA data to the paper would add important information, especially because a part of the rodent studies is based on SCFA concentrations in the gut.

2. The authors have not showed data on NAFLD in the human cohort despite the main message of the paper is that the interaction between dietary lipids and NAFLD is important. Certainly, liver biopsies are the gold standard, but any non-invasive test would be sufficient.

3. Figure 1A shows the observed species in the human cohorts, with some individuals have 5000 bacterial species. I doubt if this is correct. Can the authors explain in the method section how many species were identified in the human samples?

4. According to the data in Figure 2, group D has the highest steatosis score and liver triglyceride content, yet the histological slices are not convincing. Can the authors explain this discrepancy?

5. Figure 2, I would suggest that the authors show at least a few hepatic portal and central zones either in the extended or main figure.

6. In the extended data Figure 2 the authors show the metabolic profile of mice fed the different diets and it appears there is no correlation between insulin and hepatic steatosis, which is not what you expect. This brings me back to the main concern. The missing link between dietary lipids, gut microbiota composition and hepatic steatosis is the alterations in the plasma metabolome. Adding data on circulating SCFA, bile acids or metabolomics would significantly improve the manuscript and add novelty.

7. There are some typos that should be addressed throughout the manuscript.

RESPONSE TO REVIEWERS

We thank the reviewers for their valuable comments on our manuscript.

We have addressed the questions in the original order. Reviewer's comments are in black, while our responses are in red. Citation with new text from the manuscript is in italics. Page numbers indicated in our responses refer to page numbers in the new version of the manuscript and may differ from the numbering in the originally submitted version. In addition to the changes in text outlined below, we have also made some minor editorial changes in the manuscript. These are indicated in the Revised Marked Copy provided.

Reviewer

#1:

This is a very interesting work describing the interaction between dietary lipids and gut microbiota in liver steatosis. Overall the preclinical studies are well performed and the results are consisting.

We thank the reviewer for the encouraging remarks on our manuscript.

I would suggest, however, that the authors include the analysis of gut permeability in the different models, given its relevance for NAFLD pathogenesis.

We have performed qPCR to determine the expression of genes encoding the tight junction components ZO-1 and occludin in ileum of mice fed MF diet and diet A-G as well as in mice transplanted with microbiota from mice fed MF or diet A. However, we did not find any differences in expression. Data are presented in Extended Data Figure 4g-h and Extended Data Figure 9.

We have added the following text to the manuscript.

(Line 194):

“We also measured expression of genes encoding the tight junction proteins zonulin (Hp) and occludin (Ocln) in ileum (Extended Data Fig. 4g-h) and....”

(Line 304):

“Expression of genes encoding zonulin and occludin in ileum (Extended Data Fig. 9) and ...”

(Line 659):

“Hp-F: GCCGCTAAGAGCACAGCA, Hp-R: TGGAGGTTCCCACTCTGA, Ocln-F: GGTTGATCCCCAGGAGGCTA, Ocln-R: TAGTCAGATGGGGGTGGAGG.”

We are aware that a thorough determination of gut integrity would require additional experiments, and we cannot rule out that diet-dependent differences may occur and that it may affect metabolism and liver status. However, a deeper exploration of the phenotype would require additional major experiments that are beyond the scope of this paper.

First, I would include in Figure 1 the PCA data that the authors mention in the text.

The PCA plots have now been included as Figure 1a.

In my opinion the authors need to include an additional cohort of patients with characterized NAFLD (either liver biopsy or imaging methods) with FFQ and microbiota analysis. This sub-analysis will further confirm the hypothesis of the work.

Liver biopsies were not obtained in the IRONMET study. However, MRI data were acquired and Fatty Liver Index score (FLI, based on waist circumference, body mass index, triglyceride levels and gamma-glutamyl-transferase) was calculated. We have now used these data to relate liver status to gut microbiota composition and consumption of dietary lipids. The results are presented in Fig 1d.

We found that SFA and PUFA, but not MUFA, were positively and significantly correlated to FLI when only obese individuals were included ($n = 85$). Similar results were obtained when all individuals in the cohort were included and when dietary lipid consumption was correlated with MRI data, but these correlations did not reach significance. We also found negative correlations between microbiota diversity (Fisher's alpha and Shannon index) and liver fat determined by both MRI and FLI.

Taken together, we show that gut microbiota diversity is linked both to dietary lipid composition and liver status. The nature of these associations is investigated in the mouse experiments.

We have added the following texts to the manuscript.

(Line 120):

"To investigate the association between dietary lipids, gut microbiota diversity and liver fat we correlated dietary SFA, MUFA and PUFA, Fisher's alpha and Shannon index with liver fat determined by Magnetic Resonance Imaging (MRI) and Fatty Liver Index (FLI) in all individuals and in a subset including only individuals with obesity ($n = 85$) (Fig. 1d). The intake of SFA and PUFA correlated positively to FLI in obese individuals, but the correlation did not reach significance in the full cohort. Both Fisher's alpha and Shannon index were negatively correlated to liver fat determined by MRI and FLI in all individuals. The correlations between Fisher's alpha and MRI and between Shannon index and FLI were also significant in the subset including only individuals with obesity."

(Line 133):

“Moreover, we find a negative correlation between gut microbiota diversity and steatosis, and positive correlation between dietary SFA and MUFA and FLI in obese individuals. ”

(Line 365):

“In the present study, we demonstrate that the amount of dietary SFA influences liver status and gut microbiota profile in humans.”

(Line 382):

“In accordance with previous reports⁷ we found that impaired liver status was associated with decreased gut microbiota diversity. We also found that increased consumption of SFA and PUFA, but not MUFA, correlated with liver fat in individuals with obesity. Dietary intervention studies have shown that SFA raises liver fat levels more than PUFA and MUFA²³ while population-based studies provide limited data on the dietary risk factors associated with steatosis²⁴. The association between dietary lipid composition and both gut microbiota structure and hepatic fat status indicates a potential role of the microbiota in mediating steatosis caused by dietary lipids.”

(Line 483):

“Liver fat determined by MRI and calculation of FLI

A 1.5 Tesla scanner (Ingenia Philips Medical Systems, Eindhoven, The Netherlands) with an 8-channel receiver-coil array was used to examine all participants. The imaging protocol comprised a three-dimensional volumetric mDixon gradient-recalled echo acquisition in the axial plane that covered the entire abdominal region. We reconstructed images of fat, water, in-phase and out-of-phase (field-of-view 230 x 190 mm, repetition time 5.9 ms, excitation time 1.8 and 4 ms, flip angle 15°, number of slices 50, voxel size 1×1×1 mm, thickness 10 mm, acquisition time 6 minutes). The first array was acquired from the diaphragm to the kidneys, and the second from the kidneys to below the pubic symphysis. Each stack was acquired in 11s breath-hold task. Images of fat and water were input into Olea Sphere 3.0 in order to calculate the fat fraction map. Two regions of interest (40 pixels) were manually performed in order to determine the mean value of the right lobe of the liver, and the average was then calculated. The regions of interest were examined while avoiding breathing-related or susceptibility-related motion artifacts.

FLI, an algorithm based on waist circumference, body mass index (BMI), triglyceride, and gamma-glutamyl-transferase (GGT) for the prediction of fatty liver, was calculated as previously been described⁴”

Figure Legend Fig 1:

” Spearman correlation of SAT, MUFA and PUFA consumption, alpha diversity (Fisher’s alpha and Shannon diversity) against liver fat determined by MRI and FLI in all subjects, and in only obese subjects (n = 85) after controlling for age and sex.”

Schoeler et al. have provide insight into how changes in dietary fatty acid composition, independent of dietary fiber, influence gut microbiota profile, host metabolism and liver steatosis. This paper uses a translational approach using human data but rather focusses on obtaining mechanistic insight using rodent models. The paper holds clinical relevance and is of importance to the field.

We thank the reviewer for the positive feedback on our manuscript.

Nevertheless, the paper has not addressed how the gut microbial alterations induced by dietary fatty acids induce hepatic steatosis. I expect that some effects are mediated or driven by microbial metabolites. Adding data on circulating plasma metabolites (SCFA, bile acids or other microbial metabolites) would significantly improve the paper.

We agree that the manuscript would benefit from including metabolites linking diet-induced changes in the gut microbiota to the observed metabolic phenotype. To address this, we have measured bile acids in *vena porta* in mice fed diet A or MF diet and in mice transplanted with microbiota from mice fed A diet and MF diet. Data are presented in a new main figure (Figure 8).

We found that mice fed diet A had higher total bile acid levels (Fig 8a) and $T\beta MCA$ levels (Fig 8b) than mice fed MF diet (Fig 8a) and that the overall bile acid composition differed between mice fed the two diets (Fig 8c). The $T\beta MCA/\omega MCA$ and $\beta MCA/\omega MCA$ ratios were significantly higher in mice fed diet A (Fig 8d). This may be attributed to a decrease in the gut microbiota's capacity to perform 6 β -epimerization (Wahlström et al. 2016, PMID: 27320064). $T\beta MCA$ is a potent antagonist of FXR (Sayin 2013, PMID: 23395169) and we find that the expression of the FXR targets *Fgf15* in the ileum (Fig 8e) and the gene encoding *Shp* in the liver (Fig 8f) were decreased in mice fed diet A.

To investigate if the observed differences in bile acids between mice fed diet A and MF were mediated by the gut microbiota, we measured bile acids in mice transplanted with microbiota from mice fed either MF or diet A. Both groups were fed MF. Total bile acid levels did not differ between the groups (Fig 8g), but mice transplanted with microbiota from mice fed diet A had a trend ($p = 0.08$) towards increased $T\beta MCA$ levels (Fig 8h), a shift in overall bile acid composition (Fig 8i) and higher $T\beta MCA/T\omega MCA$ and $\beta MCA/\omega MCA$ ratios (Fig 8j). The expression of *Fgf15* did not differ significantly between the groups (Fig 8k). However, similar to the diet experiment *Shp* (Fig 8l) differ significantly between the groups.

Taken together, diet A and MF gave rise to major differences in bile acid levels and expression of FXR targets. These differences were partly transmitted by microbiota transfer. The decreased effect size in the transfer experiment may be caused by incomplete transfer of bacteria and/or by the influence of MF diet that is given to both groups. Our results show that diet-induced changes in gut microbiota composition can influence the capacity for bile acid processing. The resulting shift in bile acid composition may affect metabolic regulation.

We have added the following texts to the manuscript:

(Line 60):

“..induced a shift in bile acid profile and”

(Line 337):

“in the levels of vena porta bile acids and hepatic expression of bile acid receptor target genes

Bile acids are key mediators of microbiome-host interaction¹⁸. To identify metabolites linking the gut microbiota to the observed metabolic phenotypes we measured vena porta bile acids in mice fed diet A and MF. We found that mice fed diet A had higher total bile acid levels and tauro-beta-muricholic acid (T β MCA) levels than mice fed MF diet (Fig. 8a,b). Multivariate analysis of variance (adonis, 9999 permutations) showed that overall bile acid composition differed between mice fed diet A or MF ($p = 0.02$; Fig. 8c) and that the difference was driven by the proportions of T β MCA, tauro-omega-muricholic acid (T ω MCA), the unconjugated forms of these bile acids β MCA and ω MCA as well as tauro-cholic acid (TCA) and cholic acid (CA). The ratio between T β MCA and its product of 6 β -epimerization T ω MCA¹⁹ as well as the ratio between the unconjugated forms of these bile acids (β MCA/ ω MCA) were higher in mice fed diet A (Fig. 8d). T β MCA is a potent antagonist of FXR²⁰ and we found that the expression of the FXR targets Fgf15 in the ileum (Fig. 8e) and Shp in the liver (Fig. 8f) were decreased in mice fed diet A.

To investigate if the differences in vena porta bile acid levels between mice fed diet A and MF were mediated by the gut microbiota, we measured bile acids in the mice transplanted with microbiota from mice fed MF or diet A. Total bile acid levels did not differ between the groups (Fig. 8g), but similar to mice fed diet A mice receiving diet A microbiota showed a non-significant increase in T β MCA levels (Fig. 8h), a shift in overall bile acid composition ($p = 0.007$; Fig. 8i) and significantly higher T β MCA/T ω MCA and β MCA/ ω MCA ratios compared to mice receiving MF microbiota (Fig 8j). The expression of Fgf15 in the ileum did not differ between the groups (Fig 8k) but hepatic Shp expression was decreased in mice transplanted with diet A microbiota (Fig 8l).

Our results show that gut microbiota from diet A directly modulates vena porta bile acid profiles and the expression of genes responsive to bile acids.”

(Line 423):

“Finally, we found that differences in vena porta bile acid in mice fed diet A and MF were largely transmitted through gut microbiota transfer. Bile acids influence host metabolism via bile acid receptors such as farnesoid X receptor (FXR) and G-protein-coupled receptor-1 (TGR5) that regulate lipid-, glucose-, steroid- and energy metabolism¹⁸. Thus, bile acids have the potential to mediate the microbiota-dependent difference in phenotype between mice fed diet A and MF. The increased ratio between substrates and products of 6 β -epimerization in mice fed diet A indicates that bacteria able to perform this transformation are reduced by diet A.”

(Line 729):

“Measurement of bile acids

Bile acids were analysed using ultra-performance liquid chromatography-tandem mass spectrometry (UPLCMS/MS) according to previous work⁷¹. Briefly, samples were extracted with methanol containing deuterated internal standards. After 10 minutes of vortex and 10 minutes of centrifugation at 20 000g, the supernatant was diluted 100 times in methanol:water [1:1]. The bile acids were separated on a C18 column (1.7 μ , 2.1 x 100mm; Kinetex, Phenomenex, USA) using water and acetonitrile as mobile phases and detected using MRM in negative mode on a QTRAP 5500 mass spectrometer (Sciex, Concord, Canada). Quantification was made using external standard curves.”

Figure Legend Fig 8:

“Figure 8: The cecal microbiota from mice fed diet A regulates vena porta bile acid levels and hepatic expression of Shp in mice fed the MF diet. Bile acid levels and expression of genes regulated by bile acids in mice fed diet A or MF diet for 9 weeks (a-f) and in mice inoculated with cecal microbiota from mice fed MF or A diet and subsequently fed MF diet for 9 weeks (g-l). a, total vena porta bile acid levels, b, vena porta levels of individual bile acids, c, redundancy analysis (RDA) plot based on relative bile acid levels in vena porta, d, T β MCA/T ω MCA and β MCA/ ω MCA ratios, e, relative ileum expression of Fgf15 and f, relative hepatic expression of Shp in mice fed diet A or MF diet. g, Total vena porta bile acid levels, h, vena porta levels of individual bile acids, i, redundancy analysis (RDA) plot based on relative bile acid levels in vena porta, j, T β MCA/T ω MCA and β MCA/ ω MCA ratios, k, relative ileum expression of Fgf15 and l, relative liver expression of Shp in mice inoculated with cecal microbiota from mice fed MF or A diet and subsequently fed MF diet.

*Panel a-f: n = 10; panel g-l: n = 15 (MF microbiota), 14 (A microbiota) except for k where n = 13 for A microbiota. *p < 0.05, **p < 0.01, ***p < 0.001 determined by Mann-Whitney U test. Data are presented as mean \pm SEM. MCA, muricholic acid; CA, cholic acid; CDCA, chenodeoxycholic acid; DCA, deoxycholic acid; UDCA, ursodeoxycholic acid; LCA, lithocholic acid; HDCA, hyodeoxycholic acid; T, taurine-conjugated species.”*

1. The paper starts by showing results from the IRONMET study (n=117) where the individuals were divided into tertiles based on SFA, MUFA or PUFA consumption. Despite showing that SFA decreases gut microbiota diversity and abundance of butyrate producing bacteria, the authors have not measured SCFA in the human cohort. Adding SCFA data to the paper would add important information, especially because a part of the rodent studies is based on SCFA concentrations in the gut.

We do agree that data on SCFA in human subjects would have been very useful. Unfortunately, this was not measured, and no material is left to perform this analysis.

2. The authors have not showed data on NAFLD in the human cohort despite the main message of the paper is that the interaction between dietary lipids and NAFLD is important. Certainly, liver biopsies are the gold standard, but any non-invasive test would be sufficient.

Liver biopsies were not obtained from the subjects in the human cohort. Instead, we have used MRI data and FLI to relate gut microbiota data in the human subjects to liver status. See our response to the last question by reviewer 1.

3. Figure 1A shows the observed species in the human cohorts, with some individuals have 5000 bacterial species. I doubt if this is correct. Can the authors explain in the method section how many species were identified in the human samples?

We acknowledge that the number of observed species is high. However, different methods for analyzing metagenome data result in different numbers of features.

Since the Fisher's alpha index, based on logarithmic series model, considers the relationship between the number of species and the number of counts within the species, and as this index is significantly and negatively associated to liver fat, we have replaced Observed species with Fisher's alpha in main Fig 1 and we show Observed species in Ex Data Fig 1. As this index is most applicable to communities that are not completely inventoried and consist of few dominant taxa with tails of rare taxa (as is the case of the human gut microbiota; Falony 2016, PMID: 27126039), our results based on Fisher's alpha should be more robust to the potential bias introduced by the method used for metagenomic analysis.

4. According to the data in Figure 2, group D has the highest steatosis score and liver triglyceride content, yet the histological slices are not convincing. Can the authors explain this discrepancy?

Diet D had the second highest steatosis score and liver TG content after MF. Determination of steatosis scores was performed blinded using several microscopic views per mouse. Steatosis scores and liver TG were highly correlated. Therefore, we believe that the steatosis score is reliable. However, we agree that the displayed view is not representative of liver status in mice fed diet D and is now replaced.

5. Figure 2, I would suggest that the authors show at least a few hepatic portal and central zones either in the extended or main figure.

In addition to the liver sections displayed in fig 2f we now show pictures displaying larger areas of the liver sections in Extended Data Fig 3. In most of these pictures we have been able to include both hepatic portal and central zones.

6. In the extended data Figure 2 the authors show the metabolic profile of mice fed the different diets and it appears there is no correlation between insulin and hepatic steatosis, which is not what you expect. This brings me back to the main concern. The missing link between dietary lipids, gut microbiota composition and hepatic steatosis is the alterations in the plasma metabolome. Adding data on circulating SCFA, bile acids or metabolomics would significantly improve the manuscript and add novelty.

Data on bile acids have been added. See response to first question by reviewer 2.

7. There are some typos that should be addressed throughout the manuscript.

We have proofread the manuscript carefully and corrected several typos.

REVIEWERS' COMMENTS

Reviewer #1 (Remarks to the Author):

1) In order to analyze gut permeability they merely check for gene expression of some genes related to the intestinal barrier. Instead, LPS or LBP could be determined in the circulation, and ideally zonulin and/or other markers could be measured in feces.

2) I understand that validation of the results in other cohorts is not possible. Anyway, I appreciate that the authors included further data about the IRONMET cohort, as they provide valuable information.

Reviewer #2 (Remarks to the Author):

The authors have addressed my main concerns and provided new data and experiments which improved the manuscript. I have no further comments.

RESPONSE TO REVIEWERS

We thank the reviewers for taking their time to go through the manuscript once again. We have addressed the questions in the original order. Reviewer's comments are in black, while our responses are in red. Citation with new text from the manuscript is in italics.

Reviewer #1 (Remarks to the Author):

1) In order to analyze gut permeability they merely check for gene expression of some genes related to the intestinal barrier. Instead, LPS or LBP could be determined in the circulation, and ideally zonulin and/or other markers could be measured in feces.

Reply: This is indeed a good suggestion. Unfortunately, we do not have serum left for performing the requested analyses. We have introduced the following sentence in the manuscript to address the limitations of our approach:

Line 426-428: Thus, even though other mechanisms such as metabolic endotoxemia may be involved, bile acids have the potential to mediate the microbiota-dependent difference in phenotype between mice fed diet A and MF.

2) I understand that validation of the results in other cohorts is not possible. Anyway, I appreciate that the authors included further data about the IRONMET cohort, as they provide valuable information.

We thank the reviewer for acknowledging the value of our analysis of liver status vs metagenome data in the IRONMET cohort.

Reviewer #2 (Remarks to the Author):

The authors have addressed my main concerns and provided new data and experiments which improved the manuscript. I have no further comments.

Again, we thank the reviewer for all valuable input on our manuscript.